# Telluriding monolayer $MoS_2$ and $WS_2$ via alkali metal scooter

Seok Joon Yun[1,2], Gang Hee Han[1], Hyun Kim[1,2], Dinh Loc Duong[1], Bong Gyu Shin[1], Jiong Zhao [3], Quoc An Vu[1,2], Jubok Lee[1,2], Seung Mi Lee[4] & Young Hee Lee [1,2,5]

The conversion of chalcogen atoms to other types in transition metal dichalcogenides has significant advantages for tuning bandgaps and constructing in-plane heterojunctions; however, difficulty arises from the conversion of sulfur or selenium to tellurium atoms owing to the low decomposition temperature of tellurides. Here, we propose the use of sodium for converting monolayer molybdenum disulfide ($MoS_2$) to molybdenum ditelluride ($MoTe_2$) under Te-rich vapors. Sodium easily anchors tellurium and reduces the exchange barrier energy by scooting the tellurium to replace sulfur. The conversion was initiated at the edges and grain boundaries of $MoS_2$, followed by complete conversion in the entire region. By controlling sodium concentration and reaction temperature of monolayer $MoS_2$, we tailored various phases such as semiconducting 2H-$MoTe_2$, metallic 1T'-$MoTe_2$, and 2H-$MoS_{2-x}Te_x$ alloys. This concept was further extended to $WS_2$. A high valley polarization of ~37% in circularly polarized photoluminescence was obtained in the monolayer $WS_{2-x}Te_x$ alloy at room temperature.

[1] Center for Integrated Nanostructure Physics, Institute for Basic Science (IBS), Suwon 16419, Republic of Korea. [2] Department of Energy Science, Sungkyunkwan University, Suwon 16419, Republic of Korea. [3] Applied Physics Department, The Hong Kong Polytechnic University, Hung Hom, Hong Kong. [4] Quantum Technology Institute, Korea Research Institute of Standards and Science, Daejeon 34113, Republic of Korea. [5] Department of Physics, Sungkyunkwan University, Suwon 16419, Republic of Korea. Correspondence and requests for materials should be addressed to Y.H.L. (email: leeyoung@skku.edu)

Two-dimensional monolayer transition metal dichalcogenides (TMdCs) have unique growth behaviors that are different from bulk. While chemical vapor deposition (CVD) is commonly introduced to grow various types of TMdCs[1–4], the conversion process is also available by the substitutional reaction of chalcogen atoms[5–7]. For example, monolayer molybdenum disulfide ($MoS_2$) is converted to molybdenum diselenide ($MoSe_2$) by a heat treatment under Se-rich conditions or vice versa[6]. This method is useful for doping, alloying, spin-orbit coupling (SOC) engineering[8], and generating hetero-interfaces by selective patterning[5].

Difficulty exists, however, in the Te-conversion case due to the low decomposition temperature of tellurides and the lower activity of Te atoms with transition metals than that of sulfur or selenium atoms. Recently, molybdenum ditelluride ($MoTe_2$) and tungsten ditelluride ($WTe_2$) have been intensively investigated due to their unique phase engineering[9] and novel physical nature of Weyl semimetals and topological insulators[10,11]. Moreover, $MoS_{2-x}Te_x$ and $WS_{2-x}Te_x$ alloys with tailored SOC will further elucidate such exotic physical phenomena. To fulfill these needs, Te conversion at low conversion temperatures should be designed.

In this work, we demonstrate efficient tellurization process for converting monolayer $MoS_2$ and $WS_2$ to their tellurides via sodium-scooter ($Na_2Te$) reaction. The $Na_2Te$ scooter acts as a telluriding catalyst in which the activation barrier height for tellurization or tellurization temperature is drastically reduced to

0.73 eV or 300 °C by introducing Na-scooter. This reduction of conversion temperature leads to lower the reaction temperature as low as 525 °C, which is below the dissociation temperature of $MoTe_2$ (700 °C), the key to stabilize the converted tellurides. Tellurization occurs primarily at the edge and grain boundaries of $MoS_2$ flakes. This leads to fabricate lateral heterostructure between $MoS_2$ and 2H-$MoTe_2$. Furthermore, we obtain $MoS_{2-x}Te_x$ alloy, 2H, and 1T'-$MoTe_2$ phase by adjusting tellurization parameters and construct a phase modulation diagram with Na-scooter concentration and telluriding temperature. The bandgap and SOC can be engineered by Te composition. The partially tellurized $WS_2$ sample reveals high valley polarization up to 37% at room temperature.

## Results

**Telluriding molybdenum disulfide by sodium-assisted tellurization process.** Figure 1a is a schematic of the sodium-assisted tellurization process for $MoS_2$. A sodium hydroxide (NaOH)-coated substrate was brought into a chamber with a CVD-grown monolayer $MoS_2$ sample and the two were placed facing each other in the tellurization zone (see Methods section and Supplementary Figure 1). The $Na_2Te$, which is most probable Na–Te compound, converted from NaOH while supplying a Te vapor (Supplementary Figure 2 and Supplementary Note 1) is scooted to the $MoS_2$ substrate to exchange Te atoms with S atoms (Fig. 1b). Other compounds such as NaTe and $NaTe_3$ were

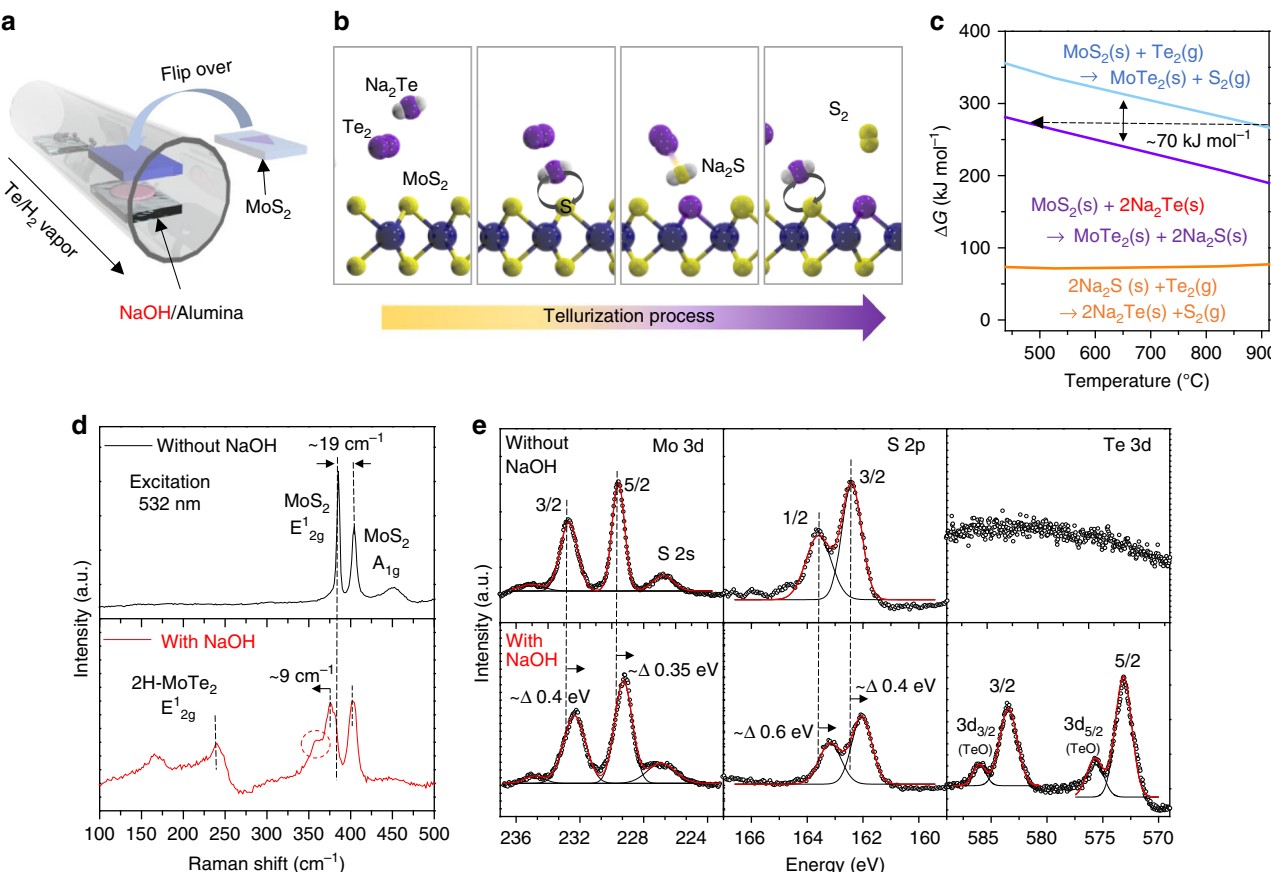

**Fig. 1** Alkali-metal-assisted conversion from $MoS_2$ to $MoTe_2$. **a** Schematic of the Na-assisted tellurization process. $MoS_2$ was flipped over to face the NaOH-coated substrate. **b** Illustration for the conversion step from Mo–S to Mo–Te via a Na-scooter. $Na_2Te$, which is regarded as the most probable Na–Te compound from sodium hydroxide, plays a role as the Te carrier and catalyst for exchanging with S atoms. After exchanging S atoms with Te atoms, $Na_2S$ is converted to $Na_2Te$ under Te-rich conditions. **c** Gibbs free energy changes for conversion without/with Na-scooter. $\Delta G$ is reduced to around 70 kJ mol$^{-1}$. **d**, **e** Raman spectra (**d**) and XPS taken for Mo, S, and Te (**e**) of partially tellurized $MoS_{2-x}Te_x$ without/with Na-scooter at 600 °C for 30 min

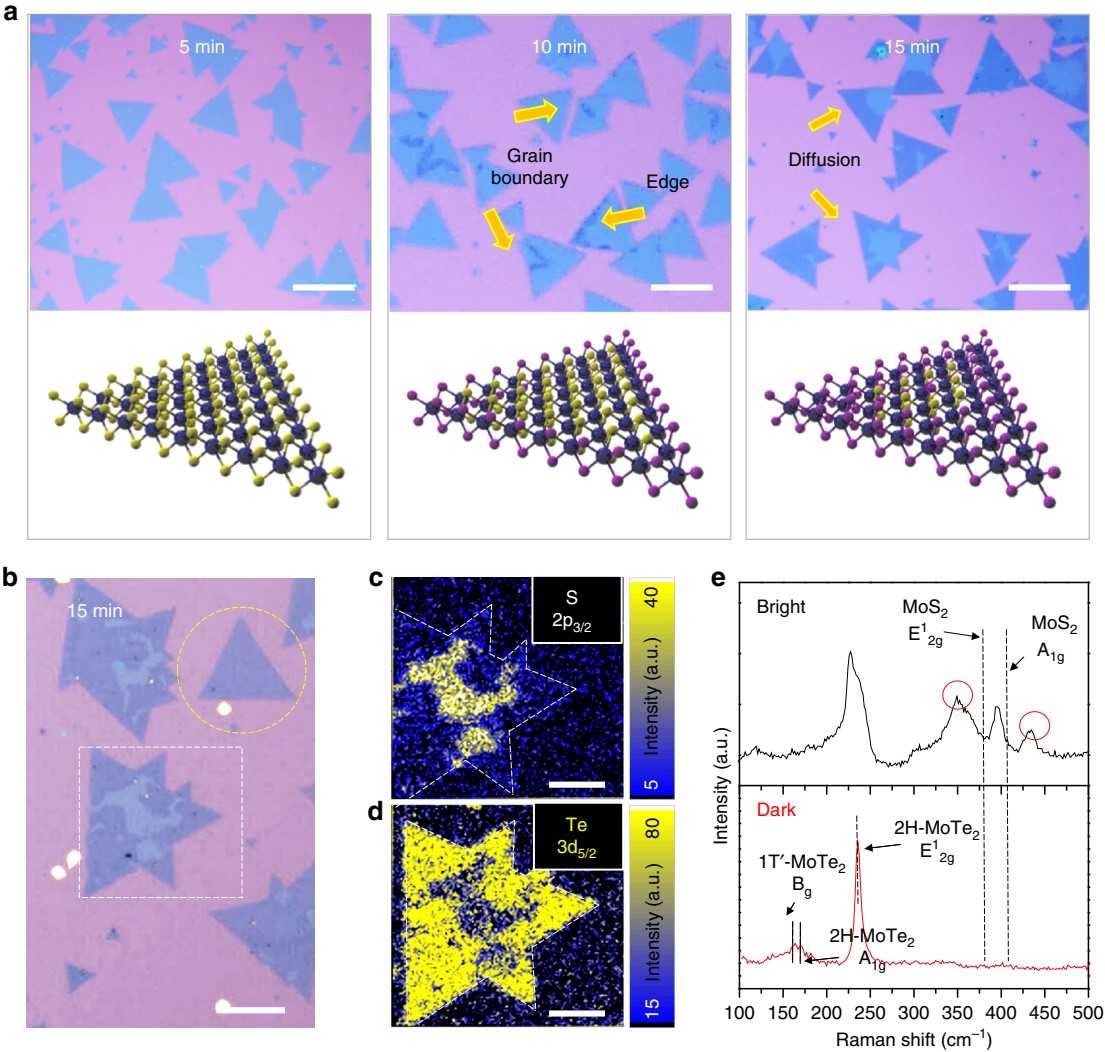

**Fig. 2** Time evolution of conversion from $MoS_2$ to $MoTe_2$. **a** Optical images and the corresponding schematics of the initial growth stages: 5, 10, and 15 min. The dark color indicates the fully converted $MoTe_2$ region from $MoS_2$. Conversion is favored at the edges and grain boundaries; scale bars, 50 μm. **b** Amplified optical image of the sample grown for 15 min. The $MoS_2$ flake is completely converted to $MoTe_2$ in the yellow-dotted circle; scale bars, 50 μm. **c**, **d** XPS mapping image of tellurized $MoS_2$ flake for S $2p_{3/2}$ (**c**) and Te $3d_{5/2}$ (**d**) in the white-dotted square in **b**; scale bars, 30 μm. A high Te content was detected in the dark region (no sulfur content was detected), while the S content (~ 20%) is seen in the bright region. **e** Raman spectra of bright and dark regions. The dark region shows the Raman spectrum exclusively from $MoTe_2$ (1T'−2H mixed phase). Some unknown peaks (349 and 435 cm$^{-1}$ in red circles) in the bright region, which are far from $MoS_2$ and $MoTe_2$ Raman peaks are observed

negligibly formed during tellurization of NaOH (Supplementary Figure 2c). In fact, $Na_2Te$ is a well-established compound as an efficient tellurizing reagent[12]. Due to its higher reactivity of $Na_2Te$ than pure Te, $Na_2Te$ is widely used for synthesizing various telluride compounds[13,14].

The reduction in the barrier height for the proposed conversion reaction via the Na-scooter ($Na_2Te$) is understood by calculating the Gibbs free energy (Fig. 1c and Supplementary Table 1). In the presence of a Na-scooter (or efficient carrier), the conversion of $MoS_2$ to $MoTe_2$ requires around 70 kJ mol$^{-1}$ less Gibbs free energy (equivalent to 300 °C or 0.73 eV by the Readhead equation[15]) than a conversion without Na. It can also be assumed that the reaction temperature at 900 °C without a Na-scooter is reduced to 500 °C in the presence of a Na-scooter (black line). This temperature reduction for tellurization is crucial to stabilize the $MoTe_2$, since the decomposition temperature of $MoTe_2$ is around 700 °C[16] (Supplementray Table 2). In this sense, Na-scooter plays as both an efficient Te carrier and catalyst. Although the Gibbs free energy is still positive, such a reaction

could occur under Te-rich conditions at a reasonable temperature. The formed $Na_2S$ could be converted back to $Na_2Te$ under Te-rich conditions (bottom-orange curve), guaranteeing continuous tellurization. There is still a trace of conversion to $MoTe_2$ even without a Na-scooter over 650 °C; however, the converted $MoTe_2$ would be etched and dissociated (Supplementary Figure 3).

The proposed mechanism is confirmed by Raman spectroscopy and X-ray photoemission spectroscopy (XPS) for the tellurized $MoS_2$ prepared with/without NaOH. The $A_{1g}$ and $E^1_{2g}$ modes of the 2H-$MoS_2$ phase observed near 402 and 383 cm$^{-1}$, respectively, in the absence of a Na-scooter, are identical to those of the pristine monolayer $MoS_2$. Meanwhile, the $E^1_{2g}$ (240 cm$^{-1}$) mode of 2H-$MoTe_2$ and some unknown peak (near 350 cm$^{-1}$ in the red-dotted circle) clearly appeared with the Na-scooter (bottom). The remaining $MoS_2$ peak is red-shifted by 9 cm$^{-1}$ for $E^1_{2g}$, indicating a compressive strain in the $MoS_2$[17] (see supplementary Figure 6 for more information). The peak shift is negligible for $A_{1g}$, indicating no appreciable charge

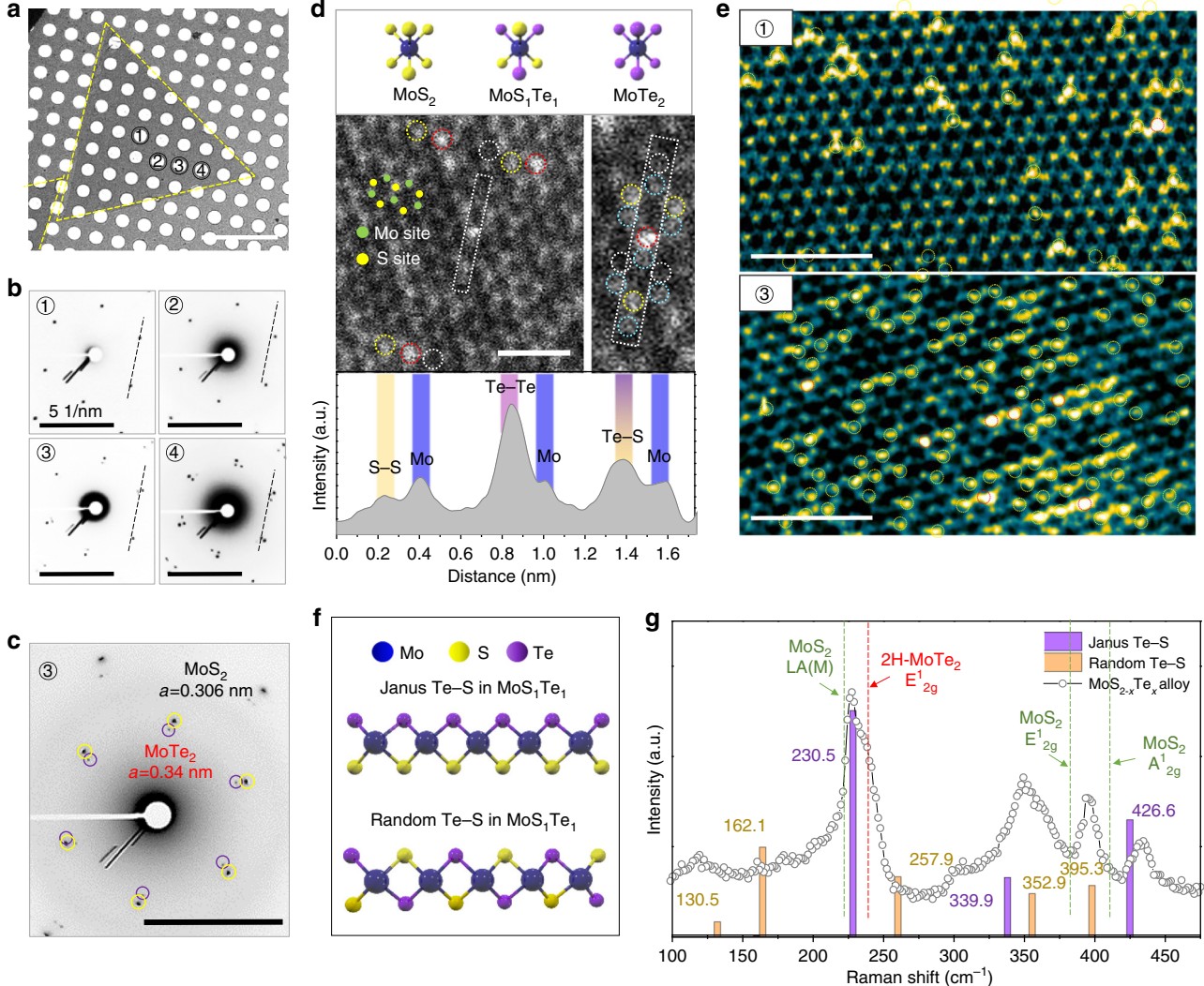

**Fig. 3** Atomic structure of monolayer $MoS_{2-x}Te_x$ with Te–S species. **a** Low-magnification TEM image of the tellurized $MoS_2$ flake on a 1.2-µm hole Cu grid; scale bar, 10 µm. **b** Electron diffraction pattern from the corresponding positions in (**a**); scale bars, 5 nm$^{-1}$. **c** Another hexagonal phase emerges from 3, revealing the $MoTe_2$ phase with a 0.340 nm lattice constant, in addition to $MoS_2$ with a lattice constant of 0.304 nm; scale bar, 5 nm$^{-1}$. **d** Schematics for three representative atomic phases of $MoS_2$, $MoS_1Te_1$ (Te–S), and $MoTe_2$. Annular dark field (ADF) STEM image with the corresponding intensity line profile of four species (white circle for S–S, yellow circle for Te–S, red circle for Te–Te, and sky-blue for the Mo site), which is distinguished by brightness; scale bar, 1 nm. **e** False-color FFT filtered images of regions 1 and 3 in **a**. Te–S and Te–Te in the chalcogen site marked by a yellow and red circle, respectively; scale bars, 2 nm. **f** Two types of Te–S arrangement in monolayer $MoS_1Te_1$ structure. **g** Analysis of Raman spectrum for $MoS_{2-x}Te_x$ alloy with calculated vibrational modes of Janus phase and randomly distributed Te–S in monolayer $MoS_1Te_1$

transfer. This implies that the obtained sample is in a form of the $MoS_{2-x}Te_x$ alloy.

More distinct features of the conversion were demonstrated from XPS analysis. While Te peaks were absent from the sample without a Na-scooter, clear Te peaks were identified with red-shifted (0.4 eV) Mo 3d peaks in the Na-assisted tellurized sample (Fig. 1e). The lowered Fermi level indicates electron withdrawal[18]. Tellurium oxide peaks were also visible, indicating that the sample is easily oxidized under ambient conditions.

**Time evolution of tellurization process**. Figure 2a displays the optical images and corresponding schematics for the time evolution of tellurization for $MoS_2$. No change in the optical contrast is seen in the sample tellurized for 5 min compared with pristine $MoS_2$. When the tellurization time is prolonged to 10 min, dark-colored regions start emerging mostly at the edges and grain boundaries. The exposed dangling bonds at the edge and some

defect sites such as grain boundaries and S vacancies are known to have higher reactivity than the basal surface of $MoS_2$. In this sense, tellurization takes place preferentially at the edge and grain boundaries in $MoS_2$. The area of the dark regions is diffused and widened from the edge to the entire area of the flakes at 15 min of tellurization (yellow-dashed circle in Fig. 2b). If the flake size is large, the longer reaction time is needed for full conversion (white-dashed box in Fig. 2b).

To analyze the chemical composition in conjunction with the optical contrast, we conducted XPS mapping for a sample tellurized for 15 min (white-dashed square, Fig. 2b). The outer region shows a relatively dark optical contrast compared to the inner region. Figure 2c and d shows XPS mapping images for S 2p$_{3/2}$ and Te 3d$_{5/2}$ (Supplementary Figure 4). S 2p$_{3/2}$ peaks were detected only in the bright region, while Te 3d$_{5/2}$ peaks were rich in the dark region. The calculated chemical compositions of Te/(S + Te) are 0.8 and 1 for the bright region and dark region,

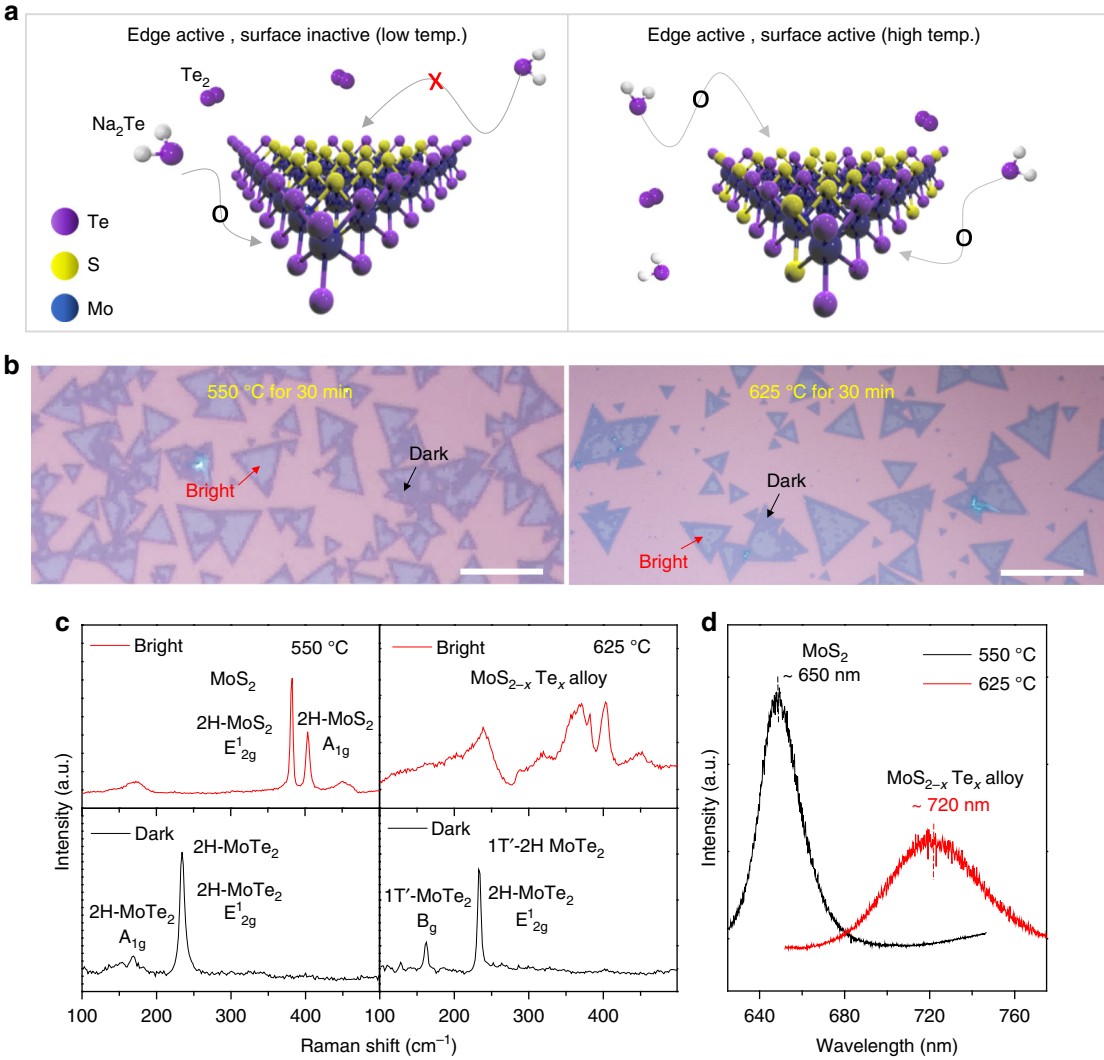

**Fig. 4** Temperature-dependent tellurization behavior of $MoS_2$. **a** Schematic for tellurization behavior of $MoS_2$ converted at low temperature and high temperature. Only the edge site of the $MoS_2$ flakes is favored over converted $MoTe_2$ at a relatively low tellurization temperature, while both the edges and surfaces of the flakes are active for tellurization at a high tellurization temperature. **b** Optical images of tellurized monolayer $MoS_2$ flakes at 550 °C and 625 °C, respectively. There are two distinct regions in terms of the optical contrast (bright and dark) in the tellurized $MoS_2$ samples; scale bars, 50 μm. **c** Raman spectra of tellurized $MoS_2$ in bright and dark regions at 550 °C and 625 °C, respectively. **d** PL spectra of the bright regions also support that negligible Te substitution happened at 550 °C. This indicates that only the edge sites of $MoS_2$ were active for tellurization at 550 °C

respectively (Supplementary Figure 5). The negligible sulfur content in the dark region indicates that $MoS_2$ is fully converted to $MoTe_2$, while the bright region still has remaining sulfur atoms.

A similar trend is also observed in the Raman spectra (Fig. 2e) The bright region reveals the $MoS_{2-x}Te_x$ alloy peaks with a $MoTe_2$-like peak near 227 $cm^{-1}$, a $MoS_2$-like peak near 395 $cm^{-1}$ and unknown peaks in red circles. To clarify the peak near 227 $cm^{-1}$, the peak was deconvoluted to LA(M) mode[19] for 2H-$MoS_2$ and $E^1_{2g}$ mode for 2H-$MoTe_2$ (more details are discussed in Fig. 3g and Supplementary Figure 6). Meanwhile, the dark region shows a uniquely $MoTe_2$ feature.

**Tellurization of molybdenum disulfide in atomic scale.** To study the tellurization of $MoS_2$ at the atomic scale, we conducted transmission electron microscopy (TEM). Figure 3a is the low-magnification TEM image of a tellurized $MoS_2$ flake grown at 625 °C and NaOH concentration of 1 μmol $cm^{-2}$ for 30 min. Electron diffraction patterns of the hexagon (Fig. 3b)

are confirmed for each region indicated in Fig. 3a. Two features are noted here: (i) at low Te loading content, $MoS_{2-x}Te_x$ alloy still maintains single crystallinity within the inner region of triangular flakes (marked by number 1, 2, and 3). (ii) Additional hexagon emergent in region 3 (expanded pattern in Fig. 3c) with a lattice constant of 0.34 nm for a lattice, equivalent to a $MoTe_2$ phase in addition to the $MoS_2$ phase ($a = 0.306$ nm), while maintaining the same orientation as $MoS_2$. When more Te content is converted (region 4) (i.e., on both sides), $MoTe_2$ may suffer from local strain, deviating the orientation of the hexagon.

Three different phases of $MoS_2$, $MoS_1Te_1$, and $MoTe_2$ are schematically drawn in the top panel (Fig. 3d). The annular dark field scanning TEM (ADF-STEM) image (middle panel) reveals three such types of intensity distributions; the brightest intensity spot in the chalcogen site corresponds to Te–Te (red circle), the middle intensity corresponds to Te–S (yellow circle), and the dark spot corresponds to S–S species (white circle). Such species were identified by comparing each intensity to the Mo intensity (bottom panel and

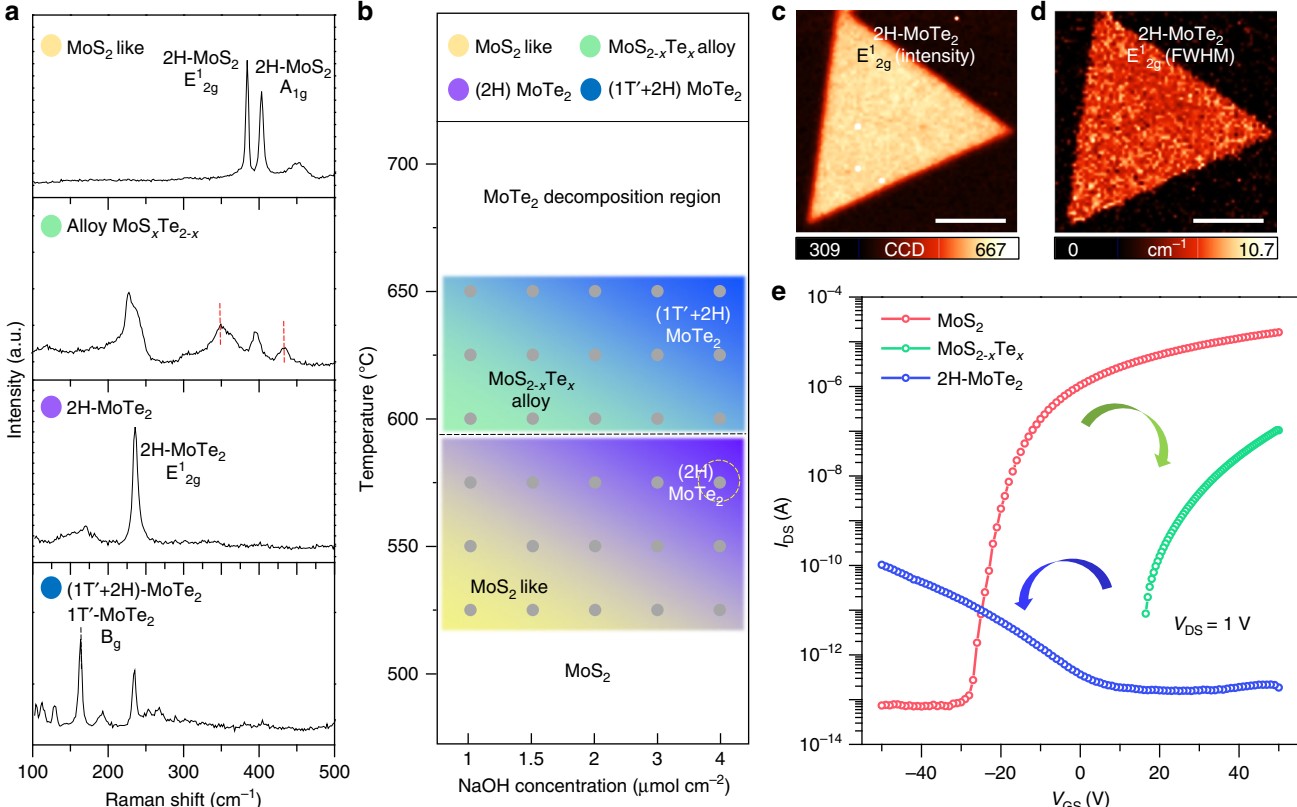

**Fig. 5** Phase modulation with NaOH concentration and temperature. **a** Raman spectra of representative $MoS_{2-x}Te_x$ phases with an excitation wavelength of 532 nm. **b** Phase modulation of $MoS_{2-x}Te_x$ identified from Raman spectra and optical images. **c**, **d** Confocal Raman mapping images for the intensity (**c**) and FWHM (**d**) of fully converted 2H-MoTe₂; scale bars, 10 μm. **e** Electrical transport characteristics of pristine MoS₂, $MoS_{2-x}Te_x$, and 2H-MoTe₂. The threshold voltage in the $MoS_xTe_{2-x}$ alloy (red) is shifted positively from pristine MoS₂ (green). The p-type behavior is shown for 2H-MoTe₂ (blue)

Supplementary Figure 7). The false-colored fast-Fourier transform (FFT) filtered image (Fig. 3e) of region 1 clearly demonstrates a few spots of Te–S and one Te–Te spot. Meanwhile, at spot 3, more Te–Te species were emergent with a large portion of Te–S species. In this case, the intensity of Mo atoms adjacent to Te atoms is exaggerated by artificial filtering (Supplementary Figure 8).

It is not clear from STEM analysis if Te atoms in $MoS_1Te_1$ are located on the top or bottom of Mo layer. There are two possible arrangements of Te–S species in $MoS_1Te_1$ structure during conversion from MoS₂ to MoTe₂ (Fig. 3f). One is called Janus phase, where S atoms at the top layer is replaced by Te atoms while retaining the bottom S layer, which is consistent with recently reported phase of $MoS_1Se_1$[20]. In addition, there is another possibility of combination, called randomly distributed phase. The total Te coverage is still half but part of Te atoms are located on top and bottom layer, distinguished from Janus phase. The corresponding Raman spectrum for $MoS_{2-x}Te_x$ alloy is provided with calculated vibrational modes of Janus phase and randomly distributed phase in Fig. 3g. Although the theoretical peaks calculated from density functional theory (DFT) represent experimental peaks to a some degree, the precise assignment of the spectra to either Janus or random phase is premature at the moment (more detail in Supplementary Figure 6). This ambiguity is partly ascribed to insufficient coverage of $MoS_1Te_1$, as shown in Fig. 3e. The intermediate phase could be an interesting phase that reveals piezoelectric properties and requires further studies. The bandgap tuning window by the tellurization of MoS₂ from 2.14 to

1.1 eV (see Supplementary Figures 9–11) is nearly twice that of $MoS_{2-x}Se_x$ alloys[21] or $Mo_{1-x}W_xS_2$ alloys[22].

**Temperature-dependent tellurization behavior.** Conversion temperature is another sensitive variable for conversion kinetics. As the tellurization temperature increases, Te content increases gradually and reaches maximum (MoTe₂) at 650 °C (Supplementary Figure 12). More interestingly, tellurization occurs more dominantly from the edge (and grain boundaries) at relatively low temperature, while this occurs on the entire surface of MoS₂ flakes at high temperature to form $MoS_{2-x}Te_x$ alloys, as shown in the schematic of Fig. 4a. Figure 4b illustrates optical images of tellurized monolayer MoS₂ flakes at two representative temperatures (550 °C and 625 °C). There are two distinct regions in terms of the optical contrast (bright and dark) in the tellurized MoS₂ samples. The corresponding Raman spectra are provided in Fig. 4c. At 550 °C, the bright regions show unaltered 2H-MoS₂ peaks, whereas the dark regions reveal semiconducting 2H-MoTe₂ peaks. At 625 °C, the bright regions reveal $MoS_{2-x}Te_x$ alloy peaks, whereas the dark regions exhibit mixed semi-conducting 2H phase and metallic 1T′ phase. This indicates that semiconducting 2H-MoTe₂ formed near the edge is favored at low temperature and metallic 1T′-MoTe₂ is favored at high temperature, reflecting the bulk phase stability[9]. Photo-luminescence (PL) is conducted for further characterization (Fig. 4d). The PL spectrum of the bright regions at 550 °C sample exhibits emission from pristine MoS₂ (650 nm), while the bright regions at 625 °C reveal an alloy peak at 1.72 eV (720 nm).

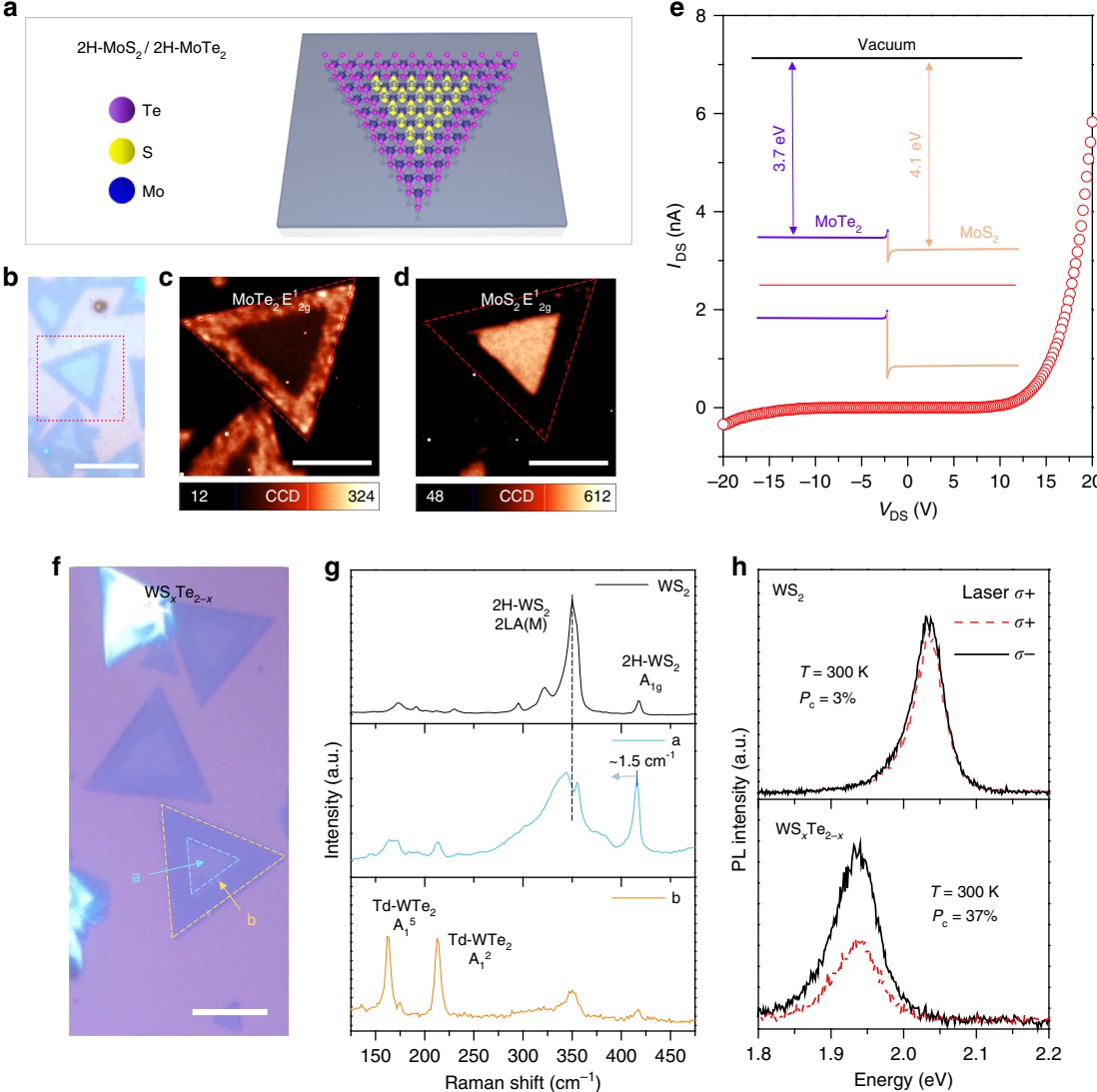

**Fig. 6** Formation of $MoS_2/MoTe_2$ lateral junction and $WS_{2-x}Te_x$ alloy. **a** Schematic of 2H-$MoS_2$/2H-$MoTe_2$ in-plane heterojunction. **b**–**e** Optical micrograph (**b**); scale bar, 20 μm, confocal Raman mapping of $E^1_{2g}$ 2H-$MoTe_2$ (**c**), and 2H-$MoS_2$ (**d**) for $MoTe_2$/$MoS_2$ junction; scale bars, 50 μm and the corresponding diode I–V curve (**e**). A clear diode behavior with the band diagram of the $MoTe_2$/$MoS_2$ heterostructure reveals a type-II junction. For electrical measurements, source ($MoTe_2$) and drain ($MoS_2$) electrodes were fabricated by depositing 5/50 nm Cr/Au. **f** Optical micrograph of tellurized monolayer $WS_2$ flakes. $WTe_2$/$WS_{2-x}Te_x$ junction; scale bar, 20 μm. **g** Raman spectra from each region of a and b in **f**. The Raman spectrum of 2H-$WS_2$ is also provided for comparison. **h** Circularly polarized PL spectra of 2H-$WS_2$ and the $WS_{2-x}Te_x$ alloy (at 300 K) excited by $\sigma+$ polarized light at energy of 2.33 eV. The PL is red-shifted to ~0.1 eV and valley polarization is enhanced up to 37% in the $WS_{2-x}Te_x$ alloy. Valley polarization ($P_c$) is defined as $P_c = \frac{I(\sigma+)-I(\sigma-)}{I(\sigma+)+I(\sigma-)}$

We also note that a sharp interface at the edge distinguished by optical contrast is formed at low temperature, while at high temperature, a mixed alloy is preferably formed. At low temperature, tellurization is preferentially initiated at the reactive sites such as the edge and grain boundaries. In that regime, more dominant enthalpic contribution than entropic contribution to Gibbs free energy results in phase segregation to minimize its interfacial energy. Meanwhile, at high temperature, the entropic contribution is dominant and Te atoms have sufficient thermal energy to overcome tellurization energy barrier randomly in the entire $MoS_2$ basal surface. This temperature-dependent behavior is also similar to the previous report on the substitution of Mo atoms in $WS_2$[23].

**Phase modulation with NaOH concentration and temperature**. To study the effect of tellurization temperature and NaOH concentration systematically, a series of experiments were done and summarized in Figs. 5a and b. Figure 5a displays the representative Raman spectra of the tellurized $MoS_2$ samples. Four representative types of phases were identified. In the top panel, a 2H-$MoS_2$-like feature is shown with two distinct $A^1_g$ and $E^1_{2g}$ peaks of $MoS_2$ and no additional peaks, although the precise positions of such peaks vary slightly with composition. The second panel shows the Raman peaks of the $MoS_{2-x}Te_x$ alloy. The third and bottom panels represent 2H-$MoTe_2$ and 2H–1T′-mixed $MoTe_2$ phases, respectively, identified by the $E^1_{2g}$ peaks of 2H-$MoTe_2$ and the $B_g$ peaks for 1T′-$MoTe_2$ (black-dotted line). Note that the 2H semiconducting phase is retained after the full conversion.

Figure 5b shows the phase modulation of $MoS_2$ with temperature and Na content determined by Raman spectra and optical images (Supplementary Figure 13). The tellurization rate increases in proportion to NaOH concentration and tellurization temperature. The conversion ratio is estimated by the area of dark region where $MoS_2$ is fully converted to $MoTe_2$ in optical images (Supplementary Figure 13a), which is expressed in the phase modulation by the gray color scale. In the temperature range from 525 °C to 575 °C, two phases of $2H\text{-}MoS_2$ and $2H\text{-}MoTe_2$ coexist due to the edge-selective tellurization. From 600 °C to 650 °C, both edge and surface are tellurized, resulting in the formation of $MoS_{2-x}Te_x$ alloy and fully converted $1T'\text{-}2H$ $MoTe_2$.

Figure 5c–d are confocal Raman mapping images of the intensity and full width at half maximum (FWHM) for $E^1_{2g}$ mode from fully converted samples of $2H\text{-}MoTe_2$ (yellow-dotted circle in Fig. 5b). The uniform contrasts of both images indicate reasonable sample uniformity while retaining a monolayer morphology without fragmentation in μm scale. The FWHM value for $E^1_{2g}$ mode is around 6.6 cm$^{-1}$ (Supplementary Figure 14), indicating that the converted $MoTe_2$ has reasonable crystallinity compared to exfoliated one (5.0 cm$^{-1}$)[24]. Tellurization was done with good uniformity in mm. The uniformity of $MoS_{2-x}Te_x$ alloy and $1T'\text{-}2H$ mixed phase $MoTe_2$ flakes are characterized by Raman mapping and the uniformity dependence on positions are also characterized by Raman spectroscopy (Supplementary Figures 15 and 16).

One interesting feature is that the proposed conversion provokes not only an alloying effect but also a carrier-type conversion. Figure 5e shows the transfer characteristics of $MoS_2$, $MoS_{2-x}Te_x$, and $2H\text{-}MoTe_2$. The pristine $MoS_2$ clearly shows an n-type behavior. With the $MoS_{2-x}Te_x$ alloy, an n-type behavior is still retained, but the threshold voltage is upshifted significantly, indicating a p-type doping effect. Completely converted $MoTe_2$ clearly demonstrates a p-type behavior; in other words, the carrier type is converted from electron to hole. It is worth mentioning that Na contamination gives n-type doping effect in $MoS_2$ FET (Supplementary Figure S17). In our case, Na compounds are washed out during transfer. Therefore, the p-type doping effect in tellurized $MoS_2$ originates exclusively from Te substitution, not from Na contamination.

**Lateral heterojunction and tellurization of $WS_2$.** We further provide a few more examples to demonstrate the strength of our efficient Na-assisted tellurization. We generate $2H\text{-}MoTe_2/2H\text{-}MoS_2$ in-plane heterojunction by telluriding $MoS_2$ at 550 °C (Fig. 6a). The optical micrograph reveals a clear contrast between the edge and inner regions (Fig. 6b). The confocal Raman mapping of the $E^1_{2g}$ peak intensity of $MoTe_2$ was identified at the edge (Fig. 6c), whereas that of $E^1_{2g}$ of $MoS_2$ was identified at the inner region (Fig. 6d). The sharp interface between $MoTe_2$ and $MoS_2$ is attributed to energy minimization by reducing interfacial energy[23], which is confirmed by the PL measurement (Supplementary Figure 18). The diode was fabricated across the interface via electron beam lithography. A clear rectifying behavior is demonstrated due to a type-II junction formation (Fig. 6e) (band diagram shown in inset).

Na-assisted tellurization is also possible for tungsten disulfide. The similar optical contrast at different regions was observed in the tellurized $WS_2$ sample (Fig. 6f). Raman spectra for regions (a) and (b) are presented with pristine $WS_2$ for comparison (Fig. 6g). While the inner region (a) reveals an alloy phase of $WS_{2-x}Te_x$, the outer region (b) clearly shows Raman modes for $T_d\text{-}WTe_2$[25]. Due to the semi-metallic nature of $T_d\text{-}WTe_2$[26], no PL was observed in

the region (b). On the other hand, the alloy phase of $WS_{2-x}Te_x$ shows a PL at 1.93 eV (2.03 eV for pristine $WS_2$) (Fig. 6h).

It has been reported that valley polarization can be tuned by doping[27], defect[28], and alloying engineering[8]. Especially, alloying can modify the valley polarization by engineered SOC. Therefore, it is intriguing to observe valley polarization of tellurized $WS_2$ by circularly polarized PL. We define the degree of valley polarization, $P_c$, as follows:

$$P_c = \frac{I(\sigma+) - I(\sigma-)}{I(\sigma+) + I(\sigma-)}, \tag{1}$$

where $I(\sigma\pm)$ is the intensity of the left (right) circularly polarized PL. No appreciable polarization was observed from pure $WS_2$ at room temperature with a 2.33 eV excitation as seen in another report[29] (top panel). On the other hand, a large valley polarization of ~ 37% is observed in the alloy which can be further enhanced by modifying the resonance radiation[30–32]. The enhancement of valley polarization can be attributed to two reasons: (i) enhanced SOC strength and (ii) inversion symmetry breaking by Te–S species. High SOC strength of Te atom can extend the spin-orbit splitting ($\Delta_{SO}$) ($\Delta_{SO}$ ($WTe_2$): 484 meV, $\Delta_{SO}$ ($WS_2$): 412 meV)[33]. We identified Te–S species in $WS_{2-x}Te_x$ alloy by STEM and simulation (Supplementary Figure 19). The Te–S species affect the out-of-plane $d_{z^2}$ orbitals and induce Rashba spin splitting. This can be another reason for enhanced valley polarization value[20].

## Discussion

In conclusion, we have realized a facile route for tellurization via a Na-scooter. The activation barrier height for conversion of $MS_2$ to $MTe_2$ (M = Mo, W) is dramatically decreased by Na-scooter, resulting in tellurization happening at relatively low temperatures where tellurides are stable. Due to emulating its host material's nature in the conversion process, we synthesized monolayer $2H\text{-}MoTe_2$ for the first time. Furthermore, the bandgap tuning window has enlarged to 1.0 eV (2.14–1.1 eV). Finally, edge-selective tellurization and modified SOC by Te content can realize two-dimensional heterojunctions and enhance the degree of valley polarization. Our approach will open up new avenues to explore intriguing physics phenomena of tellurides.

## Methods

**Synthesis of monolayer $MoS_2$ and $WS_2$ by CVD.** Monolayer $MoS_2$ and $WS_2$ were grown on $SiO_2/Si$ wafers by atmospheric pressure CVD. A water-soluble metal precursor was coated on the $SiO_2/Si$ substrate first. The precursor solution was prepared by mixing three types of chemical solutions (defined as A, B, and C).

A (Metal precursor): 0.1 g of ammonium heptamolybdate (Sigma-Aldrich, 431346) and 0.2 g of ammonium metatungstate (Sigma-Aldrich, 463922), were dissolved in 10 mL of deionized (DI) water for Mo and W precursors, respectively.

B (Promoter): 0.3 g of sodium cholate hydrate dissolved in DI water (Sigma-Aldrich, C6445) was introduced for promoting monolayer TMdCs.

C (Medium solution): An OptiPrep density gradient medium (Sigma-Aldrich, D1556, 60% (w v$^{-1}$) solution of iodixanol in water) was adopted as a medium. It does not affect the growth of TMdCs but allows for a better spin-casting process due to its high viscosity.

The precursor solution in which A, B, and C were mixed with the ratios of 2, 6, and 1, respectively, was coated onto the $SiO_2/Si$ wafer by spin casting at 3000 r.p.m. for 1 min. The precursor-coated substrate and 0.2 g of sulfur (Sigma, 344621) were separately introduced to a two-zone furnace. The sulfur zone was heated up to 210 °C at a rate of 50 °C min$^{-1}$. The substrate zone was set to 800 °C. Temperature profiles in both cases ($MoS_2$ and $WS_2$ growth) were the same, but 5 sccm of hydrogen gas was introduced only for the $WS_2$ case with 500 sccm of nitrogen as a carrier gas.

**Sodium-assisted conversion process.** NaOH of 0.1 g was dissolved in 60 mL of DI water. This NaOH solution of 25–100 μL was dropped on a 1 × 1 cm$^2$ sapphire substrate and dried in an oven (NaOH concentrations were varied from 1.0–4.0 μmol cm$^2$). The NaOH-coated substrate and $MoS_2$ on the $SiO_2/Si$ wafer were placed on the ceramic crucible (L 1.5 × W 5 × H 1.5 cm$^{-3}$). The distance between the $MoS_2$ and NaOH-coated substrate was fixed to 2 mm by placing the

NaOH substrate 1 cm above the supporter, in which the $MoS_2$ surface faced the NaOH substrate directly. A 1-inch two-zone CVD was introduced for controlling the temperature of the Te supply zone and tellurization zone, separately. The loading mass of Te is fixed to 1 g and temperature of the Te supply zone is fixed to 600 °C during the whole experiments, giving rise to a Te vapor pressure of ~ 5.91 Torr[34]. The tellurization process was conducted at atmospheric pressure with a flow rate of 500 sccm for $N_2$ gas and 25 sccm for $H_2$ gas. It should be noted that the $H_2$ gas was introduced to suppress oxidation by oxygen leakage and enhance the detachment of sulfur efficiently by the hydrodesulfurization process.

**Scanning photoelectron microscopy characterization**. The scanning photo-electron microscopy measurement for tellurized $MoS_2$ on $SiO_2$ (300 nm)/Si was performed at the 8A1 beamline of the Pohang Accelerator Laboratory in Korea. The photon source was a U6.8 undulator, and the photon energy was set to 690 eV (photon energy resolution: ~ 100 meV). With the use of the Fresnel zone, the incident X-ray beam size was ~ 200 nm. The X-rays were incident in the vertical direction to the sample surface (0°), and an electron analyzer (PHI 3057, Physical Electronics) was fixed at 54° from the incident light. This geometry allowed for the collection of sensitive information about the sample surface using a shallow probing depth. The defined energy window was 0.75 eV.

**TEM and specimen preparation**. TEM and ADF-STEM images were taken by a probe aberration-corrected JEM ARM 200 F machine, operated at 80 kV for high-resolution TEM measurements. We set the imaging time within 1 min under a high-magnification STEM node to avoid beam damage on the monolayer TMdC samples. For transferring the samples to the TEM grid, poly(methyl methacrylate) (PMMA C4, MicroChem) was coated onto the tellurized sample on the $SiO_2$/Si wafer, which was then immersed into diluted hydrofluoric acid for detaching monolayer TMdCs from the wafer by etching silicon oxide. The PMMA-supported sample was transferred to the TEM grid (PELCO, 200 mesh, copper, 1.2-μm holes) and then the PMMA was removed by dipping it into acetone for 5 min. The grid was then annealed at 180 °C in a high-vacuum chamber at a pressure of ~$7.5 \times 10^{-5}$ Torr for 12 h prior to the TEM analysis to avoid polymerization during STEM imaging.

**Electronic structures of $MoS_{2-x}Te_x$ alloy**. Scanning tunneling microscopy/spectroscopy (STM/S) was performed to investigate the electronic structure in the tellurized $MoS_2$ sample at room temperature using commercial STM (Omicron, Germany). During the measurement, the pressure in an ultra-high-vacuum chamber was below $5 \times 10^{-11}$ Torr. For STS measurements, a conventional lock-in technique was applied with a 0.05-$V_{rms}$ voltage modulation at 817 Hz. The sample was thermally cleaned below 300 °C for 1 day.

**Simulation methods**. The band structure of $MoS_{2-x}Te_x$ alloy was calculated by density functional theory with the Quantum Espresso code using a $3 \times 3 \times 1$ supercell. The exchange correlation was estimated by using the generalized gradient approximation under the Perdew–Burke–Ernzerhof functional. Projector augmented wave potentials were used for all calculations, with 80 Ry used as the cut-off energy. The structures were optimized until the pressure was lower than 0.5 kbar within the force convergence of 0.001 Ry/Bohr. An $8 \times 8 \times 1$ Monkhorst–Pack grid was used for the optimization process and the density of states was calculated with a smearing value of 0.005 Ry.

For the Raman calculations of Janus and random phase $MoS_1Te_1$, We performed quantum mechanical calculations within DFT framework. Atomic orbital basis sets were used as implemented in DMOL3 code. All electrons, including those from the core part, were considered during calculations. The exchange-correlation functions obtained using a local density approximation and the $k$-points with Monkhorst–Pack grid with a separation of 0.02/Å were used. The geometry optimization criteria were 0.005 Å for distance, 0.001 Ha/Å for force, and $10^{-5}$ Ha for total energy difference. The vibrational frequencies were computed by diagonalization of the mass-weighted second-derivative Cartesian matrix. The Hessian elements were computed by displacing each atom, and then by computing a gradient vector, thus building up a complete second-derivative matrix. In this manner, the vibrational modes were calculated numerically. The Hessian was evaluated using a two-point difference in order to reduce numerical rounding errors. The intensities were obtained from the atomic polar tensor, which was a second derivative of the total energy with respect to the Cartesian coordinates and dipole moments. The intensity of each mode was evaluated as a square of all the transition moments of the mode and expressed in terms of the atomic polar tensor matrix and eigenvectors of the mass-weighted Hessian. The displacement step size was 0.005 Å.

**Device fabrication and measurement**. The tellurized $MoS_2$ was transferred onto a highly p-doped silicon substrate with a 300-nm-thick oxide. The metal electrodes for the probe contact were patterned on the samples ($MoS_2$, $MoTe_2$, $MoS_xTe_{2-x}$, and $MoS_2/MoTe_2$ junction) by e-beam lithography followed by e-beam deposition of Cr/Au (5/50 nm). All electrical measurements were performed in high vacuum (~ $10^{-6}$ Torr) using a Keithley 4200 SCS system.

**Circularly polarized PL**. Optical measurements were performed with a lab-made laser confocal microscope at room temperature. The laser light was focused with an objective lens (100×, N/A0.85, Nikon) and the lateral resolution was estimated to be ~ 500 nm. Samples were excited with a 532-nm continuous-wave laser polarized with positive helicity ($\sigma+$) and negative helicity ($\sigma-$). Emitted light was collected with the same objective lens and guided to the spectrometer for spectral analysis. The PL spectra were analyzed for positive ($\sigma+$) and negative helicity ($\sigma-$) using a combination of a quarter-wave plate and linear polarizer placed before the spectrometer entrance slit.

**Data availability**. The data that support the findings of this study are available from the corresponding author upon request.

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

## Acknowledgements

This work was supported by the Institute for Basic Science of Korea (IBS-R011-D1) and the Human Resources Development Program (No. 20124010203270) of the Korea Institute of Energy Technology Evaluation and Planning (KETEP) grant funded by the Korean Government Ministry of Trade, Industry, and Energy.

## Author contributions

S.J.Y. designed and developed the work. J.Z. performed the TEM measurement. B.G.S. performed the STS measurement. D.D.L. calculated the density of states of the monolayer TMdCs. G.H.H. and H.K. assisted with the growth of materials. Q.A.V. performed electrical transport measurements. J.L. contributed the circularly polarized PL measurement. S.M.L. calculated the vibrational modes of MoS$_1$Te$_1$ structures. Y.H.L. guided and analyzed the work. S.J.Y. and Y.H.L. wrote the manuscript. All authors participated in the manuscript review.

## Additional information

**Competing interests:** The authors declare no competing financial interests.

