## [Peer Review File · Nature Communications]

Reviewers' comments:

Reviewer #1 (Remarks to the Author):

In this work, Seok Joon Yun etc. report the conversion of monolayer MoS₂ to MoTe₂ under a Te-rich vapor. Sodium metal is used to deliver Te atoms in their experiment. Meanwhile, they find that this method can be used to convert WS₂ to WS_{2-x}Tex. The Sodium metal anchors Te atoms and reduces the exchange barrier energy by scooting the Te atoms to replace the S atoms easily. Different growth conditions have been studied. Especially, by controlling the sodium concentration and the reaction temperature, 2H MoTe₂, metallic 1T'-MoTe₂, and 2H-MoS_{2-x}Tex alloys are obtained. This method opens a new way to the synthesis of TMD alloys and heterostructures. I recommend it to publish on nature communication after minor revisions:

- 1) Using this method, the authors can synthesize 2H MoTe₂, metallic 1T'-MoTe₂, 2H-MoS_{2-x}Tex and 2H-WTexS_{2-x}, can the author directly convert WS₂ to WTe₂?
- 2) Is it possible to produce MoS₂ and 1T'- MoTe₂ heterostructure?
- 3) Tellurization of MoS₂ will introduce strain to the lattice of MoS₂ due to the lattice mismatch of MoS₂ and MoTe₂, and thus break the MoS₂ into small pieces. The authors should elaborate the strain effect.
- 4) The author mentioned that "...During the alloying of MoS_{2-x}Tex, two cases are possible: i) top and bottom sites are replaced (Te-Te or S-S) and ii) top sites are saturated first (Te-S). Can the author shows how to they confirm this?
- 5) The authors mentioned that the high valley polarization of ~37% in circularly polarized photoluminescence was obtained in the monolayer WS_{2-x}Tex alloy at room temperature. Could the author shows the atomic ratio of S and Te and elaborate why the high valley polarization can be found in this ratio?

Reviewer #2 (Remarks to the Author):

The manuscript by Yun et al report a way to tellurize monolayer MS₂ in the presence of NaOH. A lot of results are presented in the manuscript. However, there are a few major issues in this manuscript so I can not recommend publication in its current form.

Since the manuscript reports a new method for tellurizing MoS₂, a comprehensive study of the growth mechanism is required. The authors provide a very brief discussion and propose the formation of Na₂Te as a carrier and catalyst for promoting the tellurization process. There is, however, no solid experimental evidence to support this conjecture. A systematic experimental and theoretical study probing into the growth mechanism is required. This should include a detailed study on the various important parameters, such as NaOH concentration, reaction temperature, growth time, Te loading, vapor pressure, on the structure and quality of the final product. This also means thorough structural characterization is required. For example, the authors propose the formation of MoSTe with Te atoms in one layer and S in the other, and present evidence from Raman spectroscopy and DFT. This evidence is not conclusive from my point of view. An in-depth study focusing on the growth

mechanism alone and with strong experimental support is more compelling than a study trying to cover a lot.

The quality and uniformity of the samples seem to be rather low and I don't see much value for practical applications. In particular, the authors don't seem to have good control over the structure and composition of the sample, and the bandgap varies from region to region over quite a large range. A discussion on the effect of possible Na contamination on the transport properties of the material should also be provided.

Regarding the uniformity of the samples, I do not quite understand how the authors generate the phase diagram in Figure 4b. The data point in Figure 4b should indicate single phase (except for the $1T' + 2H$ MoTe₂ regions), but the results in Figure 2 and Figure 3 seem to contradict with this. Either the data interpretation or the presentation is wrong.

Minor points:

1. Page 3, "MoS₂ could be presumably converted to MoTe₂ even without a Na-scooter over 650 °C". this sentence is rather confusing.
2. Figure 2c. the color scale bars are wrong.
3. Figure 3. The SAD in b(4) clearly shows MoTe₂ domains with a rotation, but the authors still claim single crystal. The STEM image shows some of the Mo atoms are brighter than others. Image simulation is required to show the origins of this contrast variation. The experimental STS spectra are presented without local composition while the DFT results are with composition. This might be misleading and I don't know why the authors present the data in this way. I do not understand the purpose of Figure 3h as well.
4. Figure caption on page 20. "without any structural deformation" is not accurate. From the results, the best can be claimed is "without structural deformation in the um range".
5. Figure S12b. The last column is very confusing. I don't know how these numbers should be read.
6. If the authors intend to talk about the enhanced valley polarization of tellurized WS₂, a theoretical explanation to the experimental results is required.

Reviewer #3 (Remarks to the Author):

See attached file with comments.

Summary:

1. **Observation:** tellurization of MoS₂ or WS₂ assisted by a catalytic reaction based on NaOH, lowering the temperature of the reaction to 500–600 °C
2. **Claim:** The lowering of the temperature of the reaction makes the alloyed materials possible since the MoTe₂ is thermally unstable at higher temperatures
3. **Observation:** The final material is wither MoS₂/MoTe₂ alloy, or pure MoTe₂
4. **Observation:** In most but not all the cases, the tellurization starts at edges or grain boundaries. It does not happen uniformly over the crystallite.
5. **Claim:** Te ization is driven from the edges and grain boundaries.
6. **Claim:** proposed model assumes the replacement of Te from the top surface. The authors suggested a model where top half of S plane is replaced first, leading to Te Mo S layered “Janus” structure. (not in line with claim 5).

Comments:

The results are of interest to the community and could be considered for publication, provided some of the comments are addresses.

- The experimental part is strong and convincing.
- The results are interesting and relevant to the community.
- The proposed mechanism for the Te S replacement is inadequate, does not fully explain the results and should be discussed in more details in the main text.

Detailed comments:

1. The proposed model does not take into account the propensity of the Te ization to take place at the edges and ground boundaries. The authors should consider a model where Te S replacement is driven by the defects (S vacancies) and strain, which could propagate along the crystal from the edges. Please comment in the main text.
2. Mechanism is poorly described. There is some discussion in SI Fig 13 about a temperature dependence between edge active and surface active, but is important point and should be discussed more in the main text. The authors should cite the recent work by Bogaert et al discussing the similar effect in Mo/W heterostructure growth (Nano Lett, doi: 10.1021/acs.nanolett.6b02057).
3. Figure 2 (line 330): “Conversion is favored at the edges and grain boundaries”. (not necessary correct)
 - a. The causation of the edges and grain boundaries on the Te S replacement is not necessary correct, since in Figure 2b, yellow circle shows a crystal without grain boundaries, that is fully converted, while nearby crystals with ground boundaries are not converted fully. The authors should consider that the effects of the edges and ground boundaries are not causation but correlation. It isn’t unreasonable to expect that defects are more concentrated near grain boundaries and edges and that these regions would then be the first to get tellurized. Please discuss in the main text.
 - b. SI Fig 11: Tellurization happens along grain boundaries in Fig 11b, but it is seemingly random in Fig 11c. Does not support the original claim.

- c. In the main text, the authors should describe in clear and unambiguous words all the possible effects of edges, grain boundaries and other defects.
4. The authors should comment and offer an explanation for the sharpness of the interface between the pure MTe₂ (M = Mo, W) ring and the pure MS₂ or alloy MTe_xS_{2-x} core (Fig 5, lines 157-162 & 169-171). They could refer to Bogear et al (Nano Letter 2016).
5. (Line 84) "The bright region reveals a MoTe₂ like peak near 227 cm⁻¹ and a MoS₂ like peak near 395 cm⁻¹."
 - a. "MoTe₂ like peak" is likely misidentified. The value of 227 cm⁻¹ is the exact value of MoS₂ LA(M) peak associated with point defects in MoS₂. See 10.1103/PhysRevB.91.195411 (Effect of disorder on Raman scattering of single layer MoS₂)
6. Line 59: "Meanwhile, the E₁ 2g (~240 cm⁻¹) mode of 2H MoTe₂ and..." See also Figure 1d and Fig 2d.
 - a. Note: H phase E peak is at 230 cm⁻¹, H phase A at 170 cm⁻¹ according to previous paper by Young Hee Lee (10.1039/C5CP01649E).
 - b. Please comment on the discrepancy from the previous published results
7. Fig 2c: Is the color scale backwards? Yellow = high intensity & blue = low intensity?

In conclusion, I believe the paper should be considered for publication, provided that the above comments are addressed.

Reviewer #1 (Remarks to the Author):

In this work, Seok Joon Yun etc. report the conversion of monolayer MoS₂ to MoTe₂ under a Te-rich vapor. Sodium metal is used to deliver Te atoms in their experiment. Meanwhile, they find that this method can be used to convert WS₂ to WS_{2-x}Te_x. The Sodium metal anchors Te atoms and reduces the exchange barrier energy by scooting the Te atoms to replace the S atoms easily. Different growth conditions have been studied. Especially, by controlling the sodium concentration and the reaction temperature, 2H-MoTe₂, metallic 1T'-MoTe₂, and 2H-MoS_{2-x}Te_x alloys are obtained. This method opens a new way to the synthesis of TMD alloys and heterostructures. I recommend it to publish on nature communication after minor revisions:

Response: We would like to thank the reviewer for reading our manuscript carefully. With such critical and valuable comments, we improved our manuscript significantly. For this, we really appreciate for the reviewer's comments and criticism.

1) Using this method, the authors can synthesize 2H-MoTe₂, metallic 1T'-MoTe₂, 2H-MoS_{2-x}Te_x and 2H-WTe_xS_{2-x}, can the author directly convert WS₂ to WTe₂?

Response: Yes, this is possible. Indeed we demonstrated this in Fig. 6f in the revised manuscript. Two regions (a and b) with different Te composition in WS_{2-x}Te_x are shown, which are distinguished by optical contrast in Fig. 6f. The region (a) is WS_{2-x}Te_x alloy and region (b) is Td-WTe₂ phase. Raman spectra in Fig. 6g at the bottom panel shows a clear Td-WTe₂ phase. This demonstrates the formation of complete conversion to Td-WTe₂ from 2H-WS₂.

2) Is it possible to produce MoS₂ and 1T'- MoTe₂ heterostructure?

Response: Thank you for this valuable comment. Heterointerface between 2H-MoS₂ and 1T'-MoTe₂ is possible in principle. In this study, we demonstrated 2H-MoS₂/2H-MoTe₂ in-plane heterointerface via edge selective tellurization. In the case of heterointerface between 2H-MoS₂ and 1T'-MoTe₂, two approaches could be possible. One approach is to anneal the sample of 2H-MoS₂/2H-MoTe₂ where high temperature 1T'-MoTe₂ could be realized while the other 2H-MoS₂ remains unchanged. Another approach is *in situ* CVD where 1T'-MoTe₂ is directly synthesized by conversion at high temperature by covering partially MoS₂ by passivation with oxide. The latter has been demonstrated in other heterostructure interface (MoS₂/MoSe₂: Nat. Commun. 6, 7749 (2015)). The former approach is currently under investigation in our group as an application. In our current approach, we demonstrate the concept of sodium scooter for efficient conversion process.

3) Tellurization of MoS₂ will introduce strain to the lattice of MoS₂ due to the lattice mismatch of MoS₂ and MoTe₂, and thus break the MoS₂ into small pieces. The authors should elaborate the strain effect.

Response: Thanks you for this valuable comment and we agree with the reviewer. There are two types of strains during growth: i) strain between MoS₂ and MoTe₂ (as pointed out by the reviewer) and ii) thermal expansion mismatches between substrate and host materials. The first effect is intrinsic and incurable during conversion process due to high lattice mismatch (9 %) between MoS₂ (a=3.18 Å) and MoTe₂ (a=3.46 Å). We observed tensile strain in MoS₂ and compressive strain in MoTe₂ in MoS_{2-x}Te_x alloy by analyzing Raman spectra (provided in supplementary Figure 6a). At page 3, we modified the related text to “The remaining MoS₂ peak is red-shifted by ~9 cm⁻¹ for E_{2g}¹, indicating a compressive strain in the MoS₂¹⁷ (See Supplementary Fig. 6 for more information). The peak shift is negligible for A_{1g}, indicating

no appreciable charge transfer.”

Besides, the additional strain effect from the substrate during heating and cooling was minimized by controlling the ramping speed of heating and cooling. The detailed cooling conditions are added in modified Supplementary Figure 1e with added figure caption. “Both zone are slowly cooled down without opening the furnace to minimize the strain between host materials and substrate.”

4) The author mentioned that "...During the alloying of $\text{MoS}_{2-x}\text{Te}_x$, two cases are possible: i) top and bottom sites are replaced (Te-Te or S-S) and ii) top sites are saturated first (Te-S). Can the author shows how to they confirm this?

Response: Thank you for this comment. The existence of Te-S (or S-Te) at chalcogen site in MoX_2 structure was confirmed by considering the intensity of atoms in STEM image, as shown in Fig. 3d. We observed the existence of Te-S (or S-Te) by considering intensity of chalcogen site in STEM image (S-S(lowest), Te-Te(highest) and Te-S(middle)). In STEM image, the intensity of atom is proportional to its atomic number. For example, Te atom (atomic number=52) is brighter than S atom (atomic number=16) in STEM image. In this regard, when we compare the intensity of chalcogen site with Mo atom, we can distinguish S-S, Te-Te and Te-S(or S-Te) atoms in chalcogen site. This is explained at the bottom of page 5. Similar analysis was done previously. See, for example, Nano Lett. 14, 442-449 (2014).

5) The authors mentioned that the high valley polarization of $\sim 37\%$ in circularly polarized photoluminescence was obtained in the monolayer $\text{WS}_{2-x}\text{Te}_x$ alloy at room temperature. Could the author shows **1)** the atomic ratio of S and Te and elaborate **2)** why the high valley polarization can be found in this ratio?

Response-1: Thank you very much to point it out. It is really important to know the atomic ratio of S and Te in $\text{WS}_{2-x}\text{Te}_x$ sample to correlate Te content and the degree of valley polarization, since the Te content is known to influence valley polarization from theory []. The Te content can be estimated roughly by counting Te atoms in STEM images. One typical image of high resolution STEM is shown below, which is newly added in supplementary Fig. 18(a-b). The Te content is estimated to around 3%. If Te content exceeds too high, it forms metallic Td- WTe_2 phase. In such a case, no valley polarization is observed.

There could be several reasons why the valley polarization is enhanced in $\text{WS}_{2-x}\text{Te}_x$ alloy: i) The enhanced spin-orbit coupling strength by Te atom and ii) the mirror symmetry breaking due to existence of Te-S species in $\text{WS}_{2-x}\text{Te}_x$ alloy:

- 1) Heavy atom has higher spin orbit coupling strength than light atom. So, Te atom should have higher spin orbit coupling strength than S atom. Spin orbit coupling (SOC) splits the spin state in TMDs and the degree of valley polarization is highly influenced by the value of spin orbit splitting. Calculated value of spin splitting of WTe_2 (484 meV) are 72 meV higher than WS_2 (412 meV), which could enhance the degree of valley polarization. (Annalen der Physik 526, 395-401 (2014)).
- 2) We observed the existence of Te-S species in $WS_{2-x}Te_x$ alloy, confirmed by STEM and STEM simulation in SI Fig. 18(c-f). All Te atoms are in a form of Te-S in chalcogen site in SI Fig. 18(b). This inversion symmetry breaking from Te-S species affects the out-of-plane d_{z^2} orbitals and induce Rashba spin splitting. This can be another reason for enhanced valley polarization value. Similar phenomenon was observed in $MoS_{2-x}Se_x$ alloy which has S-Se species. (Lu et al. "Janus monolayers of transition metal dichalcogenides." Nature Nanotechnology (2017).)

We added more explanations at page 10, “The enhancement of the valley polarization can be attributed to two reasons: i) enhanced SOC strength and ii) inversion symmetry breaking by Te-S species. High SOC strength of Te atom can extend the spin-orbit splitting (ΔSO) (ΔSO (WTe_2): 484 meV, ΔSO (WS_2): 412 meV)³⁴. We identified Te-S species in $WS_{2-x}Te_x$ alloy by STEM and simulation (Supplementary Fig. 18). The Te-S species affect the out-of-plane d_{z^2} orbitals and induce Rashba spin splitting. This can be another reason for enhanced valley polarization value²⁰.”

Reviewer #2 (Remarks to the Author):

The manuscript by Yun et al report a way to tellurize monolayer MS_2 in the presence of NaOH . A lot of results are presented in the manuscript. However, there are a few major issues in this manuscript so I cannot recommend publication in its current form.

Response: We appreciate for many invaluable comments. We considered your valuable comments carefully. Overall, our manuscript was improved a lot with more logical explanations. We provide here one by one responses to your comments. Although the reviewer feels that some discussions, for example, Janus phase, are not necessary in addressing our main issues of conversion process, we believe this is necessary since this phase is an intermediate phase during conversion, which strengthens our conversion process eventually. We hope that the review understand this point. If the reviewer really considers it to be mandatory to remove this part, we will remove it later.

- Since the manuscript reports a new method for tellurizing MoS_2 , a comprehensive study of the growth mechanism is required. The authors provide a very brief discussion and propose the formation of Na_2Te as a carrier and catalyst for promoting the tellurization process. There is, however, no solid experimental evidence to support this conjecture. A systematic experimental and theoretical study probing into the growth mechanism is required.

Response: Thank you for this valuable comment. It is not clear to us whether reviewer want us to clarify the formation of Na_2Te or Na_2Te as a carrier and catalyst. We tried to answer both concepts.

In our study, we proposed that the key component of Na-assisted tellurization process from MoS_2 is the formation of Na_2Te . There could be other types of stable compounds such as NaTe or NaTe_3 . In the previous manuscript, we admit that we did not describe contributions from other components. In the revised manuscript (Supplementary Figure 2c was newly added), we took XRD of our NaOH sample after completing tellurization (Please see right figure). Na_2Te peak is dominant, while other peaks of NaTe and NaTe_3 are negligible. This proves that our proposal of Na_2Te as a primary species for reaction is reasonable.

The catalytic effect of Na_2Te is in fact well known compound in the literature as an efficient tellurizing reagent (Tetrahedron 61, 7, 1613-1679 (2005)). Due to its higher reactivity of Na_2Te than pure Te , Na_2Te is widely used for synthesizing various telluride compounds. (Journal of Ovonic Research 5, 2 (2009)); (The Journal of Organic Chemistry 58, 241-244 (1993)). The formation of Na_2Te can efficiently telluride MoS_2 by reducing activation barrier height. This is understood by the reduction of Gibbs free energy, calculated in Figure 1c. The Gibbs free energy is reduced by ~ 70 kJ/mol, which is equivalent to the reduction of growth temperature by ~ 300 °C or reduction of activation barrier height by 0.73 eV, evaluated by Redhead equation, as mentioned at page 3 in the main text. This gain is critical to stabilize the product MoTe_2 . This simply suggests that Na_2Te plays a role of catalyst. Since Te is supplied via the formation of Na_2Te , Na_2Te acts as a carrier of Te gas.

For more information, we added more sentences for the the previous use of Na₂Te for efficient tellurizing reagent in the main text, page 2. “Other compounds such as NaTe and NaTe₃ were negligibly formed during tellurization of NaOH (Supplementary Fig. 2c). In fact, Na₂Te is a well-established compound as an efficient tellurizing reagent¹². Due to its higher reactivity of Na₂Te than pure Te, Na₂Te is widely used for synthesizing various telluride compounds^{13, 14}.”

- This should include a detailed study on the various important parameters, such as NaOH concentration, reaction temperature, growth time, Te loading, vapor pressure, on the structure and quality of the final product.

Response: We agree that there are many variables to test for conversion process. Some of them are interrelated as well. In our experimental conditions, there are several variables: NaOH concentration, reaction temperature, Te loading, growth time, flow rate of carrier gas, mixing ratio of carrier gas to hydrogen gas, and distance between NaOH substrate and MoS₂ substrate.

Since there are too many variables to work with and for data presentation, we chose in our study, NaOH concentration, reaction temperature and time. Other conditions are pretested and fixed. For example, the flow rate of carrier gas is also related to Te supply. Too low flow rate randomizes Te gas diffusion even to opposite direction. Too high flow rate dilute the vapor pressure of Te gas. We found that for a given 1 gram of Te loading and 600 °C vaporization temperature, 500 sccm of nitrogen gas was reasonable enough to prevent reverse flow and to reach uniform conversion condition over the entire surface of MoS₂. Small addition of hydrogen gas, in our case 25 sccm with 500 sccm of N₂ gas was necessary to promote conversion. Too small hydrogen content slowed down the conversion process and too high hydrogen content unstabilized the final product of MoTe₂. The distance between NaOH substrate and MoS₂ substrate is also related to NaOH content. The growth temperature is also related to stability of MoTe₂ phase which is well explained in the main manuscript. While most of these are described in the Method and supplementary Fig. 1 with figure caption, we added some missing part at page 13, “A 1-inch two-zone CVD was introduced for controlling the temperature of the Te supply zone and tellurization zone, independently. “The loading mass of Te is fixed to 1 gram and temperature of the Te supply zone is fixed to 600 °C during the whole experiments, giving rise to a Te vapor pressure of ~5.91 torr³⁵.” and at figure caption Fig. S1, “This is also related to NaOH content.”

With this elaborated process, we did experiment systematically depending on those experimental parameters (NaOH concentration, Tellurization temperature and time) and the effect of those parameter are discussed in Figure 2 (tellurization time), newly added Figure 4 (tellurization temperature) and modified supplementary Figure 12 and 13 (tellurization temperature vs. NaOH concentration).

- This also means thorough structural characterization is required. For example, the authors propose the formation of MoSTe with Te atoms in one layer and S in the other, and present evidence from Raman spectroscopy and DFT. 1) This evidence is not conclusive from my point of view. An in-depth study focusing on the 2) growth mechanism alone and with strong experimental support is more compelling than a study trying to cover a lot.

Response: We appreciate for this comment. We did change and rearrange the data quite a lot to focus on growth mechanism more thoroughly. For this purpose, 1) we inserted Raman data for intermediate phase and DFT calculational results to see the agreement with experimentally observed Raman peaks in Figure 3 and moved original STS data to supplementary Figure 9. 2) We also inserted new Figure 4 (temperature-dependent edge and

surface growth with schematic, optical images, Raman, and PL). 3) Furthermore we newly added kinetics study with optical images as a function of tellurization temperature and NaOH concentration in supplementary Figure 13. 4) Phase diagram of Figure 5 (previously figure 4) was modified to remove ambiguity between phases. 5) Figure 5d was newly added to demonstrate crystalline uniformity of the flake with Confocal Raman mappings. For each item, we provide more explanations below.

1) The Janus phase of MoS_1Te_1 , is an intermediate phase that occur in the middle of conversion from MoS_2 to MoTe_2 . This was clearly observed by the contrast difference of 2S(2Te) positions in STEM observations in the main manuscript. To focus more on the formation of Janus phase, Figure 3 was newly constructed with Raman data and DFT calculations (STS data was shifted to supplementary Figure 9). The related text was added at page 6, “The formation of intermediate phase of MoS_1Te_1 during conversion from MoS_2 to MoTe_2 , called Janus phase, is consistent with recently reported phase of MoS_1Se_1 ²⁰. While the replacement of top S atoms to Te atoms is kinetically more favorable in the conversion process, it is not clear from STEM analysis if Te atom in MoS_1Te_1 is located on the top or bottom of Mo layer. The corresponding Raman spectra are provided in Fig. 3g. The deconvoluted peaks (blue color) are in good agreements with DFT calculations of MoS_1Te_1 (Fig. 3f). The intermediate phase could be an interesting phase that reveals piezoelectric properties and requires further studies. The bandgap tuning window by the tellurization of MoS_2 from 2.14 eV to 1.1 eV (See supplementary figure 9-11) is nearly twice that of $\text{MoS}_{2-x}\text{Se}_x$ ²¹ or $\text{Mo}_{1-x}\text{W}_x\text{S}_2$ alloys²².”

2) To elaborate tellurization mechanism with temperature, we added new main figure 4 which was previously in supplementary information (temperature-dependent tellurization behavior). (This was also mentioned by the reviewer 3). We added the related paragraph at page 6-7, “Conversion temperature is another sensitive variable for conversion kinetics. As the tellurization temperature increases, Te content increases gradually and reaches maximum (MoTe_2) at 650 °C (Supplementary Fig. 12). More interestingly, tellurization occurs more dominantly from the edge (and grain boundaries) at relatively low temperature, while this occurs in the entire surface of MoS_2 flakes at high temperature to form $\text{MoS}_{2-x}\text{Te}_x$ alloys, as shown in the schematic of Figure 4a. Figure 4b illustrates optical images of tellurized monolayer MoS_2 flakes at two representative temperatures (550 and 625 °C). There are two distinct regions in terms of the optical contrast (bright and dark) in the tellurized MoS_2 samples. The corresponding Raman spectra are provided in Figure 4c. At 550 °C, the bright regions show unaltered 2H- MoS_2 peaks, whereas the dark regions reveal semiconducting 2H- MoTe_2 peaks. At 625 °C, the bright regions reveal $\text{MoS}_{2-x}\text{Te}_x$ alloy peaks, whereas the dark regions exhibit mixed semiconducting 2H- and metallic 1T'-phase. This indicates that semiconducting 2H- MoTe_2 formed near the edge is favored at low temperature and metallic 1T'- MoTe_2 is favored at high temperature, reflecting the bulk phase stability⁹. Photoluminescence (PL) is conducted for further characterization (Fig. 4d). The PL spectrum of the bright regions at 550 °C sample exhibits emission from pristine MoS_2 (650 nm), while the bright regions at 625 °C reveal an alloy peak at 1.72 eV (720 nm).

We also note that a sharp interface at the edge distinguished by optical contrast is formed at low temperature, while at high temperature, a mixed alloy is preferably formed. At low

temperature, tellurization is preferentially initiated at the reactive sites such as the edge and grain boundaries. In that regime, more dominant enthalpic contribution than entropic contribution to Gibbs free energy results in phase segregation to minimize its interfacial energy. Whereas at high temperature, the entropic contribution is dominant and Te atoms have sufficient thermal energy to overcome tellurization energy barrier randomly on the entire MoS₂ basal surface. This temperature-dependent behavior is also similar to previous report on the substitution of Mo atoms in WS₂.²³”

3) In supplementary Figure 13, we newly added kinetics study with optical images as a function of tellurization temperature and NaOH concentration. These are mentioned in (newly designed) Figure 5 at page 7, “Figure 5b shows the tellurization phase diagram of MoS₂ with temperature and Na content determined by Raman spectra and optical images (See supplementary Figure 13). The tellurization rate increases in proportion to NaOH concentration and tellurization temperature. The conversion ratio is estimated by the area of dark region where MoS₂ are fully converted to MoTe₂ in optical images (Supplementary Fig. 13a), which is expressed in the phase diagram by the grey color scale. In the temperature range from 525 °C to 575 °C, two phases of 2H-MoS₂ and 2H-MoTe₂ coexist due to the edge selective tellurization. From 600 °C to 650 °C, both edge and surface are tellurized, resulting in the formation of MoS_{2-x}Te_x alloy and fully converted 1T'-2H MoTe₂.”

4) Phase diagram of Figure 5 (previously figure 4) was modified to remove ambiguity between phases. As mentioned in the previous paragraph, the conversion ratio is estimated by the area of dark region where MoS₂ are fully converted to MoTe₂ in optical images (Supplementary Fig. 13a), which is expressed in the phase diagram by the grey color scale. Therefore the previous abrupt phase transformation was smoothed in the modified figure.

5) The sample uniformity of the flake with Confocal Raman mappings after conversion was demonstrated by new figure 5c,d and supplementary Fig. 14 and 15. We added more explanation at page 8, “Figures 5c–d are confocal Raman mapping images of the intensity and full width at half maximum (FWHM) for E_{2g}¹ mode from fully converted samples of 2H-MoTe₂ (yellow dotted circle in Fig. 5b). The uniform contrasts of both images indicate reasonable sample uniformity while retaining a monolayer morphology without fragmentation in μm scale. The FWHM value for E_{2g}¹ mode is around 6.6 cm⁻¹ (Supplementary Fig. 14), indicating that the converted MoTe₂ has reasonable crystallinity compared to exfoliated one (5.0 cm⁻¹)²⁴. Tellurization was done with good uniformity in mm scale. (Supplementary Fig. 14f-h). The uniformity of MoS_{2-x}Te_x alloy and 1T’-2H mixed phase MoTe₂ flakes are also characterized by Raman mapping and shows reasonable

uniformity (Supplementary Fig. 15).”

• The quality and uniformity of the samples seem to be rather low and I don't see much value for practical applications.

Response: For the concern of quality and uniformity of the samples, we conducted confocal Raman mapping for 1) fully converted 2H-MoTe₂, 2) MoS_{2-x}Te_x alloy and 3) mixed 1T'-2H phase MoTe₂. These are newly added supplementary figure 14 and 15.

1) In the case of fully converted 2H-MoTe₂, we provided Raman mapping image for intensity (Fig. S14c) and full with half maximum (FWHM) (Fig. S14d) of E_{2g}¹ mode from 2H-MoTe₂. The uniform contrast in those images indicates the uniformity of the sample. The extracted FWHM value of E_{2g}¹ mode is around 6.6 cm⁻¹ indicates that the converted 2H-MoTe₂ has reasonable crystallinity compared to the value of exfoliated one (5.0 cm⁻¹). (Nano lett. 14, 11 6231-6236 (2014))

The tellurization occurred uniformly in mm scale: pristine MoS₂ (Fig. S14f), fully converted 2H-MoTe₂ (Fig. S14g) and fully converted 2H-MoTe₂ in mm scale (Fig. S14h)

In this regard, we believe 2H-MoTe₂ converted from 2H-MoS₂ has reasonable crystallinity and uniformity. We emphasize the uniformity and crystallinity of fully converted 2H-MoTe₂ by putting FWHM mapping image in Figure 5d and added these crystallinity and uniformity issues in the revised manuscript page 8. “Figures 5c–d are confocal Raman mapping images of the intensity and full width at half maximum (FWHM) for E_{2g}¹ mode from fully converted samples of 2H-MoTe₂ (yellow dotted circle in Fig. 5b). The uniform contrasts of both images indicate reasonable sample uniformity while retaining a monolayer morphology without fragmentation in μm scale. The FWHM value for E_{2g}¹ mode is around 6.6 cm⁻¹ (Supplementary Fig. 14), indicating that the converted MoTe₂ has reasonable crystallinity compared to exfoliated one (5.0 cm⁻¹)²⁴. Tellurization was done with good uniformity in mm scale. (Supplementary Fig. 14f-h). The uniformity of MoS_{2-x}Te_x alloy and 1T'-2H mixed phase MoTe₂ flakes are also characterized by Raman mapping and shows reasonable uniformity (Supplementary Fig. 15).”

To demonstrate the strength of our approach, we grow MoS_{2-x}Te_x alloy and mixed 1T'-2H phase MoTe₂ by tellurizing MoS₂ over 600 °C and their uniformity are characterized by confocal Raman mapping and also added newly in supplementary figure 15. Fig. S15a-d are

optical image (a), Raman mapping images of alloy peaks for $345\sim 385\text{ cm}^{-1}$ (b), $435\sim 475\text{ cm}^{-1}$ (c) and its representative Raman spectra (d) marked in (a) (please see below). The uniform contrast in Raman mapping images guarantee uniformity of $\text{MoS}_{2-x}\text{Te}_x$ alloy in micrometer scale. ($625\text{ }^\circ\text{C}$ and $1\text{ }\mu\text{mol cm}^{-2}$ for 30 min.).

Fig. S15e-h are optical image (e), Raman mapping image for E_{2g}^1 mode of 2H-MoTe_2 (f), B_g mode of $1\text{T}'\text{-MoTe}_2$ (g) and representative Raman spectra (h) marked in regions (a) (see below). At high temperature, Te desorption vigorously occurs, resulting in Te deficient $1\text{T}'$ phase MoTe_2 . Although we are not sure why $1\text{T}'$ -phase are generated in the middle of the flake, it seems that each phases are segregated to each other to minimize interfacial energy. The single phase of $1\text{T}'\text{-MoTe}_2$ is observed (3 and 5 points in Fig. S15f). This implies that there is a chance to grow fully converted $1\text{T}'\text{-MoTe}_2$. We added some more sentences as mentioned above.

- In particular, the authors don't seem to have good control over the structure and composition of the sample, and the bandgap varies from region to region over quite a large range.

Response: For concerning structure (phase) and composition controls, several figures are added as described earlier in the above questions and modified accordingly. For example, structural phase evolution with growth temperature are described in new Figure 4 and supplementary figure 13. Abrupt phase changes with temperature and NaOH concentration was smoothed by color which represents smooth modulation of phases in new Figure 5b. Compositions are also modulated with growth temperature and evaluated by XPS (supplementary Figure 12). Te composition gradually increases as the growth temperature increases.

To see the bandgap variation by Te substitution in monolayer MoS_2 , we used $\text{MoS}_{2-x}\text{Te}_x$ alloy sample for STS measurement. It should be noted that all STS spectra were extracted in $5\text{ }\text{\AA} \times 5\text{ }\text{\AA}$ spot size (limited by tip size). Within that scale, there could be numerous configurations of $\text{MoS}_{2-x}\text{Te}_x$ alloy sample, as shown in STEM image of $\text{MoS}_{2-x}\text{Te}_x$ alloy. There are many configurations in $5\text{ }\text{\AA} \times 5\text{ }\text{\AA}$ scale such as MoS_2 (white box), less Te doped MoS_2 (yellow box) and MoTe_2 like (blue box). Therefore, the bandgap extracted varies largely with positions. Nevertheless, for a given temperature and NaOH concentration, the sample uniformity is maintained in a

micrometer scale, distinguished by optical contrast in optical microscope (supplementary Figure 13). For a complete conversion, the sample uniformity is proven by confocal Raman mappings (Fig. 5c,d and supplementary Figure 14,15).

- A discussion on the effect of possible Na contamination on the transport properties of the material should also be provided.

Response: Thank you for your valuable comment. To confirm the effect of Na contamination to electrical transport of MoS₂ sample although we believe that most Na compounds are water soluble and easily removed during transfer, we fabricated the nine set of MoS₂ FETs and measured electrical transport before and after NaOH doping and now newly added in supplementary Figure 16 (right for your convenience).

It is clear in the figure that Na-contamination gives n-type doping effect to MoS₂. In this regard, we exclude the possibility of n-doping by Na in our case and Na compounds are washed out during transfer. We therefore conclude that the p-type doped tellurized MoS₂ in our sample originates from Te substitution. See another reference for n-doping effect of MoS₂ by Na. (Applied Physics Letters 105, 24 241602 (2014)). We commented the effect of Na contamination for electrical transport of MoS₂ in the main text. “It is worth mentioning that Na contamination gives n-type doping effect on MoS₂ FET (Supplementary Fig. S16). In our case Na compounds are washed out during transfer. Therefore, the p-type doping effect in tellurized MoS₂ originates exclusively from Te substitution, not from Na contamination.”

- Regarding the uniformity of the samples, I do not quite understand how the authors generate the phase diagram in Figure 4b. The data point in Figure 4b should indicate single phase (except for the 1T'+2H MoTe₂ regions), but the results in Figure 2 and Figure 3 seem to contradict with this. Either the data interpretation or the presentation is wrong.

Response: Thank you for your valuable comment. To avoid confusion, we added more experimental data for optical images for the tellurization kinetics with growth temperature and NaOH concentration (supplementary Fig. 13). We modified accordingly the phase diagram in Figure 5b (right for your reference). The smooth color represents the area ratio of two phases at the boundary, which was roughly extracted from the area ratio from optical micrographs in the supplementary Fig. 13. We modified the related text at page 8, “Figure 5b shows the tellurization phase diagram of MoS₂ with temperature and Na content determined by Raman spectra and optical images (See supplementary Figure 13). The tellurization rate increases in proportion to NaOH concentration and tellurization temperature. The conversion ratio is estimated by the area of dark region where MoS₂ are fully converted to MoTe₂ in optical images (Supplementary Fig. 13a), which is expressed in the phase diagram by the grey color scale. In the temperature range from 525 °C to 575 °C, two phases

of 2H-MoS₂ and 2H-MoTe₂ coexist due to the edge selective tellurization. From 600° to 650 °C, both edge and surface are tellurized, resulting in the formation of MoS_{2-x}Te_x alloy and fully converted 1T'-2H MoTe₂.”

Supplementary Figure 13

Minor points:

1. Page 3, “MoS₂ could be presumably converted to MoTe₂ even without a Na-scooter over 650 °C”. this sentence is rather confusing.

Response: We have tried to tellurize MoS₂ sample without Na-scooter at various temperature (625 °C, 650 °C, 675 °C and 700 °C) (Supplementary Figure 3). Although the tellurized MoS₂ sample over 675 °C show Raman mode of E¹_{2g} for 2H-MoTe₂, the structure of tellurized MoS₂ flakes are broken at this temperature because this temperature is too close to the decomposition temperature of 2H-MoTe₂ (700 °C). In this regard, we concluded that monolayer MoS₂ cannot be converted to MoTe₂ with retaining its monolayer structure without the presence of NaOH. To avoid this confusion, we changed the sentence to in page 3, “There is still a trace of conversion to MoTe₂ even without a Na-scooter over 650 °C; however, the converted MoTe₂ would be etched and dissociated (Supplementary Fig. 3).”

2. Figure 2c. the color scale bars are wrong.

Response: Thank you very much. We modified it accordingly.

3. Figure 3. The SAD in b(4) clearly shows MoTe_2 domains with a rotation, but the authors still claim single crystal.

Response: Thank you for your valuable comment. It is well known that triangular shape of MoS_2 flakes are single-crystalline. With this hypothesis, we assume tellurized MoS_2 flakes are also single-crystalline because the substitution of Te atom should not distort the structure of the host material. This hypothesis seems reasonable that in Figure 3c, both MoS_2 and MoTe_2 peaks appear with the same orientation. However, we realize that the structure can be distorted at high Te loading due to applied strain from lattice constant mismatch between MoTe_2 and MoS_2 or 1T' phase transition in Fig. 3b(4). In this regard, we modified our manuscript at page 5, “(i) At low Te loading content, $\text{MoS}_{2-x}\text{Te}_x$ alloy still maintains single crystallinity within the inner region of the triangular flakes (marked by number 1, 2 and 3).”

3-1. The STEM image shows some of the Mo atoms are brighter than others. Image simulation is required to show the origins of this contrast variation.

Response: Thank you for this careful concern. To show its origin, we did STEM simulation and added it in new supplementary Fig. 8. During FFT filtering process for clarifying STEM image, we masked diffraction patterns of $\text{MoS}_{2-x}\text{Te}_x$ a bit larger to show its structure clearly. However, the increase of the mask size can affect point spread function (PSF) value in STEM image. (Similar behavior when the spot size of electron beam is widen). To see the effect of PSF value, we simulated Te-doped MoS_2 structure depending on the PSF width. As you can see in Fig. S8b, the intensity of Mo atoms (marked in red circles) which are adjacent to Te atom (marked in yellow circle) becomes higher when the PSF width increases. Because of this, the intensity of Mo atoms adjacent to Te atoms is exaggerated. We added one sentence at page 6, “In this case, the intensity of Mo atoms adjacent to Te atoms is exaggerated by artificial filtering (See supplementary Figure 8).”

3-2 The experimental STS spectra are presented without local composition while the DFT results are with composition. This might be misleading and I don't know why the authors present the data in this way. I do not understand the purpose of Figure 3h as well.

Response: Thank you for thoughtful comment. The reviewer is right to see this point. We wanted to interpret STS data from various points. Since we do not know information of the composition of each point experimentally, we did DFT calculations for bandgaps with various compositions to see if there is a correlation between two values. Although both did show some correlation, we admit that this is not a rigorous approach as pointed out by the reviewer.

Therefore, we rearranged the DFT calculation data (previously Figure 3g and h) into modified supplementary Figure 11 where DFT calculation of Janus phase and non-Janus phase are presented. This is separated from experimental STS data set (supplementary figure 9) for readers. We leave the detailed interpretation to future works.

4. Figure caption on page 20. “without any structural deformation” is not accurate. From the results, the best can be claimed is “without structural deformation in the μm range”.

Response: Thank you very much for this careful comment. We modified it as recommended in the revised supplementary figure 14b because the AFM figure and caption are moved to supplementary figure 14 from the main figure 4c. “The uniform optical contrast in (a) and AFM image (b) show no significant structural deformation in the μm range.”

5. Figure S12b. The last column is very confusing. I don’t know how these numbers should be read.

Response: Thank you for pointing it out and sorry for our confusing description. The last column Te composition in $\text{MoS}_{2-x}\text{Te}_x$ alloy. We changed it to x ($\text{MoS}_{2-x}\text{Te}_x$).

b

Tellurization Temp. ($^{\circ}\text{C}$)	Mo 3d (eV)		S 2p (eV)		Te 3d (eV)		x ($\text{MoS}_{2-x}\text{Te}_x$)
	3/2	5/2	1/2	3/2	3/2	5/2	
525	233.11	230.03	162.85	164.04	-	-	0
550	233.20	230.06	162.77	163.94	584.53	574.15	0.33
575	232.76	229.72	163.61	162.49	583.87	573.51	0.54
600	232.40	229.25	163.49	162.27	583.78	573.43	0.87
625	231.90	228.73	163.83	162.68	583.05	572.67	1.03
650	231.00	227.88	-	-	582.95	572.55	2

6. If the authors intend to talk about the enhanced valley polarization of tellurized WS_2 , a theoretical explanation to the experimental results is required.

Response: Thank you very much to point it out. We provided information for the atomic ratio of S to Te in $\text{WS}_{2-x}\text{Te}_x$ sample to correlate Te content and the degree of valley polarization. The Te content can be estimated roughly by counting Te atoms in STEM images. One typical image of high resolution STEM is shown below, which is newly added in supplementary Fig. 18(a-b). The Te content is estimated to around 3 %.

We assume that there are mainly two reasons why the valley polarization is enhanced in $\text{WS}_{2-x}\text{Te}_x$ alloy. 1) The enhanced spin-orbit coupling strength by Te atom and 2) the mirror symmetry breaking due to existence of Te-S species in $\text{WS}_{2-x}\text{Te}_x$ alloy.

1) Heavy atom has higher spin-orbit coupling strength than light atom. So, Te atom should have higher spin-orbit coupling strength than S atom. Spin-orbit coupling (SOC) splits the

spin state in TMDs and the degree of valley polarization are highly influenced by the value of spin-orbit splitting. The calculated value of spin splitting of WTe_2 (484 meV) is higher by 72 meV than WS_2 (412 meV), which could enhance the degree of valley polarization. (Annalen der Physik 526, 395-401 (2014))

- 2) We observed the existence of Te-S species in $\text{WS}_{2-x}\text{Te}_x$ alloy, which is confirmed by STEM measurement and STEM simulation in SI Fig. 15(c-f). All Te atoms exist as Te-S in chalcogen site in SI Fig. 15(b). This inversion symmetry breaking from Te-S species affects the out-of-plane d_{z^2} orbitals and induces Rashba spin splitting. This can be another reason for enhanced valley polarization value. Similar phenomenon was observed in $\text{MoS}_{2-x}\text{Se}_x$ alloy which has S-Se species. (Lu et al. "Janus monolayers of transition metal dichalcogenides." Nature Nanotechnology (2017).)

We added these theoretical explanation in the revised manuscript at page 10, “The enhancement of the valley polarization can be attributed to two reasons: i) enhanced SOC strength and ii) inversion symmetry breaking by Te-S species. High SOC strength of Te atom can extend the spin-orbit splitting (Δ_{SO}) ($\Delta_{\text{SO}}(\text{WTe}_2)$: 484 meV, $\Delta_{\text{SO}}(\text{WS}_2)$: 412 meV)³⁴. We identified Te-S species in $\text{WS}_{2-x}\text{Te}_x$ alloy by STEM and simulation (Supplementary Fig. 18). The Te-S species affect the out-of-plane d_{z^2} orbitals and induce Rashba spin splitting. This can be another reason for enhanced valley polarization value²⁰.”

Reviewer #3 (Remarks to the Author):

Summary:

1. **Observation:** tellurization of MoS₂ or WS₂ assisted by a catalytic reaction based on NaOH, lowering the temperature of the reaction to 500 ~ 600 °C
2. **Claim:** The lowering of the temperature of the reaction makes the alloyed materials possible since the MoTe₂ is thermally unstable at higher temperatures
3. **Observation:** The final material is wither MoS₂/MoTe₂ alloy, or pure MTe₂
4. **Observation:** In most but not all the cases, the tellurization starts at edges or grain boundaries. It does not happen uniformly over the crystallite.
5. **Claim:** Tellurization is driven from the edges and grain boundaries.
6. **Claim:** proposed model assumes the replacement of Te from the top surface. The authors suggested a model where top half of S plane is replaced first, leading to Te-Mo-S layered "Janus" structure. (not in line with claim 5).

Comments:

The results are of interest to the community and could be considered for publication, provided

some of the comments are addresses.

- The experimental part is strong and convincing.
- The results are interesting and relevant to the community.
- The proposed mechanism for the Te S replacement is inadequate, does not fully explain the results and should be discussed in more details in the main text

Response: We really appreciate for these valuable comments.

For concerning 5 and 6, we observed edge selective growth and claimed top half S plane is first replaced to Te due to favorable exchange kinetics, which is reasonable. More precisely from experimental observations, Te substitution occurs from the edge and complete the substitution at the edge and move inside later.

Other reviewers also asked to explain tellurization mechanism more rigorously with experimental data. Your suggestions and the introduced reference are really helpful to support growth mechanism.

Detailed comments:

1. The proposed model does not take into account the propensity of the Tellurization to take place at the edges and ground boundaries. The authors should consider a model where Te S replacement is driven by the defects (S vacancies) and strain, which could propagate along the crystal from the edges. Please comment in the main text.

Response: Thank you for your valuable comment. We should mentioned where and why is tellurization preferred at the edge and grain boundaries in MoS₂ flake. We added the sentence for that in revised manuscript page 4. "The exposed dangling bonds at the edge and some defect sites such as grain boundaries and S vacancies are known to have higher reactivity than basal surface of MoS₂. In this sense, tellurization takes place preferentially at the edge and grain boundaries in MoS₂."

2. Mechanism is poorly described. There is some discussion in SI Fig 13 about a temperature dependence between edge active and surface active, but is important point and should be discussed more in the main text. The authors should cite the recent work by Bogaert et al

discussing the similar effect in Mo/W heterostructure growth (Nano Lett, doi: 10.1021/acs.nanolett.6b02057).

Response: We appreciate for this comment and introducing a suitable reference for this. The recent work for metal substitution reported by Bogaert et al was very helpful to support the behavior of chalcogen substitution in $\text{MoS}_{2-x}\text{Te}_x$ alloy. We cited this work and discussed our observation based on this in newly made Figure 4 (page 7). “We also note that a sharp interface at the edge distinguished by optical contrast is formed at low temperature, while at high temperature, a mixed alloy is preferably formed. At low temperature, tellurization is preferentially initiated at the reactive sites such as the edge and grain boundaries. In that regime, more dominant enthalpic contribution than entropic to Gibbs free energy results in phase segregation to minimize its interfacial energy. Whereas at high temperature, the entropic contribution is dominant and Te atoms have sufficient thermal energy to overcome tellurization energy barrier randomly on the entire MoS_2 basal surface. This temperature-dependent behavior is also similar to previous report on the substitution of Mo atoms in WS_2 .”

In addition, we modified substantially the main manuscript and supplementary to strengthen the growth mechanism. To summarize them, i) more concrete evidence of Na_2Te formation with additional XRD (Supplementary 2C), ii) referring to previous reports on Na_2Te as a source for tellurization, iii) rearrangement of Figure 3 to focus more on intermediate phase (Janus) to explain growth mechanism (STEM, Raman, DFT) by moving STS data to supplementary, iv) new figure 4 for growth mechanism with temperature dependence, v) new figure supplementary Fig. 13 for kinetics study with optical images of tellurization as a function of tellurization temperature and NaOH concentration, and iv) modification of Figure 5b (phase diagram) to elaborate growth kinetics and Figure 5c,d (confocal Raman mappings) to demonstrate the sample uniformity.

3. Figure 2 (line 330): “Conversion is favored at the edges and grain boundaries”. (not necessary correct)

a. The causation of the edges and grain boundaries on the Te S replacement is not necessary correct, since in Figure 2b, yellow circle shows a crystal without grain boundaries, that is fully converted, while nearby crystals with grain boundaries are not converted fully. The authors should consider that the effects of the edges and grain boundaries are not causation but correlation. It isn't unreasonable to expect that defects are more concentrated near grain boundaries and edges and that these regions would then be the first to get tellurized. Please discuss in the main text.

Response: Thank you for your valuable comment. It is clear from all of our experiments that tellurization occurs at the edge and grain boundaries first particularly at low temperature (Fig. 2a and supplementary Figure 13). In this case, it takes time to diffuse into the inside the flake if the flake size is large. On the other hand, as pointed out by the reviewer, Fig. 2b (See right) shows two flakes: with smaller flake but no grain boundary the flake is fully tellurized (yellow circle) and with grain boundaries but large-size flake some inner part is still not tellurized (white square). The diffusion rate of tellurization should be similar to all flakes and it is reasonable to regard that larger flake needs longer time than smaller one.

To demonstrate the size dependence, we also added another optical image here (See left).

Both of flakes (marked by circles) are single crystalline. The large

flake (yellow circle) has not fully tellurized whereas small flake (red circle) is fully converted. In comparison single crystalline (yellow circle) with poly crystalline (white circle), poly crystalline flake seems to be more tellurized than single crystalline although it is not clear. We added this modification at page 4, “In this sense, tellurization takes place preferentially at the edge and grain boundaries in MoS₂. The area of the dark regions is diffused and widened from the edge to the entire area of the flakes at 15 min of tellurization (yellow-dashed circle in Fig. 2b). If the flake size is large, longer reaction time is needed for full conversion (white-dashed box in Fig. 2b).”

b. SI Fig 11: Tellurization happens along grain boundaries in Fig11b, but it is seemingly random in Fig 11c. Does not support the original claim.

Response: Thank you very much for reading our manuscript carefully. Tellurization is initiated at the grain boundaries in both images but the direction of tellurization diffusion is random. The yellow circles are initiation points but diffuse randomly. In the revised manuscript, we provided a more comprehensive set of optical images as a function of temperature and NaOH concentration in supplementary Figure 13.

c. In the main text, the authors should describe in clear and unambiguous word all the possible the effect of edges, grain boundaries and other defects.

Response: In the revised manuscript, we modified the related sentences for the effect of edge, grain boundaries and sulfur vacancy in figure 2 content. Please look for red color for the changed parts.

4. The authors should comment and offer an explanation for the sharpness of the interface between the pure MTe₂ (M = Mo, W) ring and the pure MS₂ or alloy MTe_xS_{2-x} core (Fig 5, lines 157 162 & 169 171). They could refer to Bogear et al (Nano Letter 2016).

Response: Thank you very much for this comment. We provided more information for sharpness of interface between 2H-MoTe₂ and 2H-MoS₂ with modified PL mapping image and its representative spectra in Supplementary Figure 17(c and f). If the junction is not abrupt, the peak shift induce by alloy phase should be observed. As you can see in Fig. S17f, the peak position is not changed indicate sharp interface between MoS₂ and MoTe₂. We mention it at page 9 and Supplementary Fig. 17.

“The sharp interface between MoTe₂ and MoS₂ is attributed to energy minimization by reducing interfacial energy²⁷, which is confirmed by the PL measurement (Supplementary Fig. 17).”

“PL is highly sensitive to doping. Inner part shows pristine MoS₂ PL spectrum. No distinguishable change in the peak position is observed across the junction, indicating that a sharp interface between MoS₂ and MoTe₂ is formed.” Supplementary Figure 17.

5. (Line 84) “The bright region reveals a MoTe₂ like peak near 227 cm⁻¹ and a MoS₂ like peak near 395 cm⁻¹.”

a. “MoTe₂ like peak” is likely misidentified. The value of 227 cm⁻¹ is the exact value of MoS₂ LA(M) peak associated with point defects in MoS₂. See 10.1103/PhysRevB.91.195411 (Effect of disorder on Raman scattering of single layer MoS₂)

Response: Thank you for your valuable comments. To clarify this, the peak near 227 cm⁻¹ is deconvoluted to three peaks. Two peaks are well matched to LA(M) mode for 2H-MoS₂ and E_{2g}¹ mode for 2H-MoTe₂. The peak near 204 cm⁻¹ is unknown at this moment. For this, supplementary Figure 6a is modified and we added a new sentence in the main manuscript at page 5, “To clarify the peak near 227 cm⁻¹, the peak was deconvoluted to LA(M) mode¹⁹ for 2H-MoS₂ and E_{2g}¹ mode for 2H-MoTe₂ (More details are discussed in Figure 3g and supplementary Fig. 6).”

6. Line 59: “Meanwhile, the E_{2g}¹ (~240 cm⁻¹) mode of 2H MoTe₂ and...” See also Figure 1d and Fig 2d.

a. Note: H phase E peak is at 230cm⁻¹, H phase A at 170cm⁻¹ according to previous paper by Young Hee Lee (10.1039/C5CP01649E).

b. Please comment on the discrepancy from the previous published results

Response: Thank you for careful reading of our manuscript. Actually, the paper mentioned above is theoretically calculated value of Raman spectrum of MoTe₂. There is another reference which shows Raman spectra of 2H-MoTe₂ depending on thickness. In their report, bulk 2H-MoTe₂ shows E_{2g}¹ peak at ~ 235 cm⁻¹ while monolayer shows E_{2g}¹ peak at ~ 236.5 cm⁻¹, which is similar to our values. We cited this paper (Yamamoto et al. ACS Nano, 8, 3895-3903 (2014))

7. Fig 2c: Is the color scale is backwards? Yellow = high intensity & blue = low intensity?

Response: Thank you very much for pointing out our mistake. We edited it accordingly.

In summary, we really appreciate for reviewer's efforts to criticism and invaluable comments. We substantially improved our manuscript accordingly.

Reviewers' Comments:

Reviewer #1:

Remarks to the Author:

I am satisfied with the revisions made by the authors, and thus recommend it to publish on Nature Comm.

Reviewer #2:

Remarks to the Author:

I appreciate the effort from the authors in revising the manuscript to address the technical comments from the reviewers. Some of my comments in the previous report have been addressed. However, there are still a few major issues left in this revised version.

1. The Janus MoSTe structure. I'm glad that the authors cite Ref 20 for the Janes structure in monolayer TMD. However, it seems like the authors may have misunderstood single Te substitution as the so called Janus structure. The STEM images do show single Te sites, but it doesn't prove that all the Te atoms are in one atomic plane while S in the other. The Raman data doesn't provide conclusive evidence as well. At least, calculated phonon modes for the randomly distributed Te structure should be provided for comparison. And the STEM images don't show a composition of 50% Te.

2. The "phase diagram" is still very confusing. Phase diagram should describe the presence of distinct phases at thermodynamic equilibrium under given well defined conditions. However, from the results and discussions, it seems to me that the "phase diagram" presented in Figure 5 only describes the structures of the samples obtained from specific growths at given temperatures and NaOH "relative" concentrations. My understanding is that by increasing the Te supply and growth time the sample should eventually convert to MoTe₂. If that's the case, the schematic in Figure 5 should not be called a phase diagram. In addition, describing the NaOH concentration in a "phase diagram" as "relatively low" and "relatively high" doesn't provide any useful information.

3. I was a bit confused why I commented on the STEM images in my previous report. Then I looked at the data in Figure 3 and Figure S7 more carefully, and realized that the image processing was misleading. In the raw data, Te sites are brighter than Mo, while the FFT filtered images show the opposite. This is certainly wrong.

4. To demonstrate the uniformity of the samples, Raman spectra taken from random positions, instead of Raman mapping, should work better as peaks from impurity phases would show up. Again my overall comment remains the same. The authors should try to take out some unnecessary data, instead of adding more, to make the paper more focused. In addition, claims should only be made when there are solid evidences either from experiment or theory.

Reviewer #3:

Remarks to the Author:

I believe the authors were successfully addressed all the comments, and that the modified manuscript is suitable for publication in the current form.

Reviewer #1 (Remarks to the Author):

I am satisfied with the revisions made by the authors, and thus recommend it to publish on Nature Comm.

Response: We would like to thank the reviewer again for accepting our manuscript to be published on Nature Comm.

Reviewer #3 (Remarks to the Author):

I believe the authors were successfully addressed all the comments, and that the modified manuscript is suitable for publication in the current form.

Response: We would like to thank the reviewer again for accepting our manuscript to be published on Nature Comm.

Reviewer #2 (Remarks to the Author):

I appreciate the effort from the authors in revising the manuscript to address the technical comments from the reviewers. Some of my comments in the previous report have been addressed. However, there are still a few major issues left in this revised version.

1. The Janus MoSTe structure. I'm glad that the authors cite Ref 20 for the Janus structure in monolayer TMD. However, it seems like the authors may have misunderstood single Te substitution as the so called Janus structure. The STEM images do show single Te sites, but it doesn't prove that all the Te atoms are in one atomic plane while S in the other. The Raman data doesn't provide conclusive evidence as well. **At least, calculated phonon modes for the randomly distributed Te structure should be provided for comparison.** And the STEM images don't show a composition of 50% Te.

Response: We agree with the reviewer that in fact there is other phase like random phase other than Janus phase in MoS_1Te_1 (in new Fig. 3f). For this concern, we added new Raman peak positions from randomly distributed MoS_1Te_1 (in new Fig. 3g and new Supplementary Fig. 6). Although the theoretical peaks calculated from DFT represent experimental peaks to some degree, the precise assignment of the spectra to either Janus or random phase is premature at the moment. This ambiguity is partly ascribed to insufficient coverage of Te-S species in $\text{MoS}_{2-x}\text{Te}_x$ as shown in Fig. 3e (lower panel). Nevertheless, more than half of the coverage is still Te-S species in $\text{MoS}_{2-x}\text{Te}_x$. This is also pointed out by the reviewer. At this moment, we leave as an open question for future. The related text in figure caption and the main manuscript (page 6) was modified accordingly. **"It is not clear from STEM analysis if Te atoms in MoS_1Te_1 are located on the top or bottom of Mo layer. There are two possible arrangements of Te-S species in MoS_1Te_1 structure during conversion from MoS_2 to MoTe_2 (Fig. 3f). One is called Janus phase, where S atoms at the top layer is replaced by Te atoms while retaining the bottom S layer, which is consistent with recently reported phase of MoS_1Se_1 ²⁰. In addition, there is another possibility of combination, called randomly distributed phase. The total Te coverage is still half but part of Te atoms are located on top and bottom layer, distinguished from Janus phase. The corresponding Raman spectrum for $\text{MoS}_{2-x}\text{Te}_x$ alloy is provided with calculated vibrational modes of Janus phase and randomly distributed phase in Fig. 3g. Although the theoretical peaks calculated from DFT represent experimental peaks to a some degree, the precise assignment of the spectra to either Janus or random phase is premature at the moment (more detail in Supplementary Fig. 6). This ambiguity is partly ascribed to insufficient coverage of MoS_1Te_1 , as shown in Fig. 3e (lower**

panel)..”

Figure 3f and g | f, Two types of Te-S arrangement in monolayer MoS_1Te_1 structure. g, Analysis of Raman spectrum for $\text{MoS}_{2-x}\text{Te}_x$ alloy with calculated vibrational modes of Janus phase and randomly distributed Te-S in monolayer MoS_1Te_1 .

Supplementary Figure 6 | Calculated phonon modes for the Janus-phase and random Te-S distributed MoS_1Te_1 structure. a, Schematic for various phonon vibration modes of the Janus-phase and random Te-S distributed MoS_1Te_1 structure. b, The comparison between deconvoluted Raman spectrum of $\text{MoS}_{2-x}\text{Te}_x$ alloy and calculated Raman modes of Janus and Random Te-S distributed MoS_1Te_1 by the local density approximation.

2. The “phase diagram” is still very confusing. Phase diagram should describe the presence of distinct phases at thermodynamic equilibrium under given well defined conditions. However, from the results and discussions, it seems to me that the “phase diagram” presented in Figure 5 only describes the structures of the samples obtained from specific growths at given temperatures and NaOH “relative” concentrations. My understanding is that by increasing the Te supply and growth time the sample should eventually convert to MoTe_2 . If that’s the case, the schematic in Figure 5 should not be called a phase diagram. In addition, describing the

NaOH concentration in a “phase diagram” as “relatively low” and “relatively high” doesn’t provide any useful information.

Response: Thank you for your concern and valuable comment. We changed “phase diagram” to “phase modulation” in the main text and figure caption. The NaOH concentration (1, 1.5, 2, 3, 4 $\mu\text{m cm}^{-2}$) is also presented in x-axis of modified Figure 5b.

3. I was a bit confused why I commented on the STEM images in my previous report. Then I looked at the data in Figure 3 and Figure S7 more carefully, and realized that the image processing was misleading. In the raw data, Te sites are brighter than Mo, while the FFT filtered images show the opposite. This is certainly wrong.

Response: Thank you for reading our manuscript carefully. Yes, it was certainly wrong. We changed the filtering method for STEM image to correct the error. We provide comparison between previous and modified STEM images for $\text{MoS}_{2-x}\text{Te}_x$ alloy here for the reviewer’s reference. For clear comparison, we marked Mo sites and Te-S sites by red dotted circles and white dotted circles, respectively. As you commented, Mo sites are shown brighter than Te-S site in previous filtered STEM image (wrong obviously). We did filtering carefully and modified STEM image, revealing good agreement with raw data in terms of brightness (Te-S sites are brighter than Mo sites). We changed the filtered STEM image in the main figure 3e and supplementary Figure 7 and 8. We appreciate this comment very much.

4. To demonstrate the uniformity of the samples, Raman spectra taken from **random positions**, instead of Raman mapping, should work better as peaks from impurity phases would show up.

Response: Thank you very much for your suggestion. As we presented in supplementary Figure 13, we have many different phases. Among them, we chose representative four conditions for MoS_2 like, 2H MoTe_2 , $\text{MoS}_{2-x}\text{Te}_x$ alloy and 1T'-2H MoTe_2 to investigate uniformity dependence on the position. We measured Raman spectroscopy at four regions which are marked by numbers for each sample (newly added in supplementary Figure 16a). All the samples show similar Raman spectra to each other except the 1T'-2H MoTe_2 case (e) where the relative composition of 1T' phase to 2H phase is slightly different from each other. We mentioned this issue at page 9. “The uniformity of $\text{MoS}_{2-x}\text{Te}_x$ alloy and 1T'-2H mixed phase MoTe_2 flakes are characterized by Raman mapping and the uniformity dependence on positions are also characterized by Raman spectroscopy (Supplementary Fig. 15 and 16).”

Supplementary Figure 16 | Sample uniformity of tellurized MoS₂ sample at different positions.

Again my overall comment remains the same. The authors should try to take out some unnecessary data, instead of adding more, to make the paper more focused. In addition, claims should only be made when there are solid evidences either from experiment or theory.

Response: We appreciate for the reviewer's valuable comments. We already deleted several contents during the first revision. Since the Nature Comm Article is rather comprehensive, we would like to keep information as much as we can to assist readers, which we believe different from Letter papers. For the concern of Janus phase, we present other possible Te-S arrangement which is called random phase with calculated Raman mode as you mentioned. We cannot say conclusively that Te-S species in MoS₂-xTex alloy exist as a Janus phase or random phase from Raman calculations. Thus, we removed any argument about assignment. However, we would like to provide information of those structures because Te-S species were frequently observed in our STEM data. Please understand our intention.

REVIEWERS' COMMENTS:

Reviewer #2 (Remarks to the Author):

I have no further technical questions, but would like to point out that the "random" MoSTe model presented in Fig 3f actually looks ordered from the side view. I still believe the discussion of MoS₁Te₁ structure is unnecessary as the images in Fig 3 do not show a 50% Te concentration. What is there is simply single Te substitution sites, not a new MoS₁Te₁ structure. But I will leave this to the editor and the authors to decide.